

# GRACE-REC: a reconstruction of climate-driven water storage changes over the last century

Vincent Humphrey[1,2], Lukas Gudmundsson[1]

[1] Institute for Atmospheric and Climate Science, ETH Zurich, Switzerland
5   [2] Division of Geological and Planetary Sciences, California Institute of Technology, Pasadena, CA, USA

*Correspondence to*: Vincent Humphrey (vincent.humphrey@env.ethz.ch)

**Abstract.**

The amount of water stored on continents is an important constraint for water mass and energy exchanges
10  in the Earth system and exhibits large inter-annual variability at both local and continental scales. From
2002 to 2017, the satellites of the Gravity Recovery and Climate Experiment mission (GRACE) have
observed changes in terrestrial water storage (TWS) with an unprecedented level of accuracy. In this
paper, we use a statistical model trained with GRACE observations to reconstruct past climate-driven
changes in TWS from historical and near real time meteorological datasets at daily and monthly scales.
Unlike most hydrological models which represent water reservoirs individually (e.g. snow, soil moisture,
etc.) and usually provide a single model run, the presented approach directly reconstructs total TWS
changes and includes hundreds of ensemble members which can be used to quantify predictive
uncertainty. We compare these data-driven TWS estimates with other independent evaluation datasets
such as the sea level budget, large-scale water balance from atmospheric reanalysis and in-situ streamflow
measurements. We find that the presented approach performs overall as well or better than a set of state-
of-the-art global hydrological models (Water Resources Reanalysis version 2). We provide reconstructed
TWS anomalies at a spatial resolution of 0.5°, at both daily and monthly scales over the period 1901 to
present, based on two different GRACE products and three different meteorological forcing datasets,
resulting in 6 reconstructed TWS datasets of 100 ensemble members each. Possible user groups and
applications include hydrological modelling and model benchmarking, sea level budget studies,
assessments of long-term changes in the frequency of droughts, the analysis of climate signals in geodetic
time series and the interpretation of the data gap between the GRACE and the GRACE Follow-On



mission. The presented dataset is publicly available (https://doi.org/10.6084/m9.figshare.7670849) and updates will be published regularly.

**Copyright statement.**

**Contents**




**1 Introduction**

Because the amount of freshwater available on land controls the development of natural ecosystems as much as human activities, terrestrial water storage (TWS) represents a critical variable of the Earth system. Changes in TWS can be caused by both anthropogenic and natural processes. Natural variability in ocean and atmospheric circulation, such as the El Niño Southern Oscillation (ENSO), is responsible for anomalies in precipitation which strongly influence water storage (Ni et al., 2017), leading to regional droughts and floods with large impacts on human activities (Veldkamp et al., 2015). At the global scale, climate-driven fluctuations in the total amount of water stored on land have been linked to a wide range of geophysical phenomena, including changes in global mean sea level (Cazenave et al., 2014;Reager et al., 2016;Rietbroek et al., 2016;Dieng et al., 2017), changes in global carbon uptake by land ecosystems (Humphrey et al., 2018), and the motion of the Earth's rotational axis (Adhikari and Ivins, 2016;Youm et al., 2017). In addition to climate-driven natural variability, human activities also influence terrestrial water storage, for instance through groundwater depletion (Rodell et al., 2009;Chen et al., 2016), building of dams (Chao et al., 2008), or the impact of anthropogenic climate change on land ice (Jacob et al., 2012).

From 2002 to 2017, changes in terrestrial water storage (TWS) have been measured by the GRACE satellites with an unprecedented accuracy. Because these observations integrate both natural and anthropogenic effects across all water reservoirs (i.e. soil moisture, groundwater, snow, lakes, wetlands, rivers and land ice), isolating the contribution of specific reservoirs or the relative importance of natural



versus anthropogenic effects is still relatively uncertain and has been the focus of several recent publications (Reager et al., 2016;Eicker et al., 2016;Wada et al., 2016;Fasullo et al., 2016;Felfelani et al., 2017;Getirana et al., 2017;Pan et al., 2017;Andrew et al., 2017;Rodell et al., 2018;Hanasaki et al., 2018;Khaki et al., 2018;Cazenave, 2018). In this context, one critical aspect is to model the effect of

climate variability on TWS changes. At this time, only global hydrological models and land surface models can provide long-term estimates of natural TWS variability, however, they are usually not calibrated against GRACE measurements and sometimes exhibit large biases in TWS amplitude (Schellekens et al., 2017;Zhang et al., 2017;Scanlon et al., 2018). Typically, only a small number of such model runs is available and exploring the uncertainty related to the use of different meteorological forcing

datasets is not possible. With this paper, we aim to address these shortcomings with a computationally cheap alternative. Unlike hydrological models which represent physical processes and model water reservoirs individually (e.g. snow, soil moisture, lakes, etc.), we train a statistical model to directly reconstruct the total TWS changes from precipitation and temperature information.

The primary objective of this paper is to provide long and consistent time series of climate-driven TWS variability. Although the temporal coverage of GRACE observations will be extended by the GRACE Follow-On mission launched on May 22 2018, there will be a temporal gap of approximately one year between the two missions. The reconstruction provided here is calibrated against GRACE measurements and can be used to interpret this data gap and reconcile the two datasets. In addition, we provide a century-

long TWS reconstruction that can be used to study past natural TWS variability. We expect that this product will be relevant to sea level budget studies (Chambers et al., 2016;Cheng et al., 2017;Frederikse et al., 2018;Cazenave, 2018), the analysis of climate signals in geodetic time series (in GRACE or in e.g. ground GNSS measurements), development of daily hydrological loading models (Dill and Dobslaw, 2013;Moreira et al., 2016), as well as global to regional assessments of the recurrence of extreme

hydrological droughts and their impact on ecosystems (Sheffield and Wood, 2007;Sheffield et al., 2012;Beguería et al., 2014;Griffin and Anchukaitis, 2014;Kusche et al., 2016;Dai and Zhao, 2016;Spinoni et al., 2017;Heim, 2017;Rudd et al., 2017;Sinha et al., 2017;Haslinger and Blöschl, 2017;Um et al.,



2017;Bento et al., 2018;D'Orangeville et al., 2018;Huang et al., 2018;Markonis et al., 2018;Anderegg et al., 2018;Gao et al., 2018).

## 2 Data and Methods

### 2.1 GRACE products

The two different monthly GRACE solutions used here (Table 1) are obtained using the so-called mass concentration (mascon) technique. This technique provides estimates of mass changes over small predefined regions, that are referred to as *mascons*. The two solutions differ in terms of the employed processing algorithms and also in terms of the models used to correct for the effect of glacial isostatic adjustement (GIA). For more general information on the GRACE mission, gravity recovery techniques

and processing, we refer the reader to the reviews of Wouters et al. (2014) or Wahr (2015).

### 2.2 Precipitation and temperature

We use three different precipitation products which are aimed to address the needs of various user communities (Table 2). The multi-source weighted-ensemble precipitation dataset (MSWEP) merges a large number of existing precipitation products, including satellite-based, raingauge-based and reanalysis

products (Beck et al., 2017;Beck et al., 2018). We expect this dataset to provide a best-estimate for the period 1979-2016. The Global Soil Wetness Project Phase 3 (GSWP3) forcing dataset (Kim, 2017) is based on the 20$^{th}$ Century Reanalysis (20CR) version 2c (Compo et al., 2011). The original 20CR precipitation fields produced at a resolution of 2° are dynamically downscaled using spectral nudging and bias-corrected using observations from the Global Precipitation Climatology Project (GPCP) and the

Climatic Research Unit (CRU). With this dataset, we aim to provide a homogeneous long-term reconstruction of climate-driven TWS changes over the period 1901-2014. Third, we use precipitation estimates from the European Centre for Medium-Range Weather Forecast (ECMWF) re-analysis (ERA-Interim), which cover the period 1979-present. With this dataset, we aim to provide frequent updates of reconstructed TWS anomalies which can, for instance, be used to investigate the data gap between the

GRACE mission (decommissioned in October 2017) and the GRACE Follow-On mission launched in May 2018. For temperature, we use ERA-Interim air temperature in combination with MSWEP and ERA-





Interim precipitation, and GSWP3 air temperature in combination with GSWP3 precipitation. We note
that sensitivity analyses have shown that the choice of the temperature dataset has very little influence on
the final product (not shown).

**2.3 Modelling approach**

**2.3.1 Model formulation**

A simple statistical model is calibrated at each GRACE mascon individually, meaning that model
parameters are space-dependent. One model is calibrated for each combination of the two GRACE
products (Table 1) with the three precipitation products (Table 2). Because the model described here does
not have any explicit constraint in terms of mass or energy conservation, we refer to it as a statistical
model, however its formulation is largely inspired from basic principles of hydrological modelling.
Assuming a linear water store model, water outputs are directly proportional to the storage and to the
residence time of the water store (e.g. Beven, 2012), so that the temporal evolution of the storage can be
approximated as:

$$\text{TWS(t)} = \big(\text{TWS(t}-1) + \text{P(t)}\big) \cdot e^{-\frac{1}{\tau(t)}} \tag{1}$$

where $t$ is a daily time vector, $TWS(t)$ is the storage, $P(t)$ is the precipitation input and $\tau(t)$ is the
residence time of the water store.

Small (large) values of the residence time indicate that water inputs tend to leave the reservoir quickly
(slowly), either through runoff or evapotranspiration. Here we introduce seasonal changes in residence
time (e.g. related to snow accumulation during the cold season or increased evaporative demand during
the warm season) using a temperature-dependent relationship. The residence time used in Eq. (1) is
formulated as a function of de-trended daily air temperature:

$$\tau(t) = a + b \cdot T_Z(t) \tag{2}$$



Where $a$ and $b$ are calibrated model parameters with positive sign and $T_Z(t)$ is a transformation of the original de-trended daily air temperature T(t). The purpose of this transformation is to first make $\tau$ only sensitive to changes in temperature when temperature is higher than 0° Celsius,

$$T_0 = \begin{cases} 0, & T < 0 \\ T, & T \geq 0 \end{cases} \tag{3}$$

and to moderate the influence of extreme temperature values by applying a sigmoid transform to the standardized temperature:

$$T_Z = 1 - \tanh\left(\frac{T_0 - \text{Mean}(T_0)}{\text{StDev}(T_0)}\right) \tag{4}$$

As a result of this transformation, $T_Z$ approaches a value of 1 (0) when temperature gets colder (warmer) and thus the residence time increases (decreases) (Eq. 2). Note that different or more complex formulations (e.g. also involving net radiation) were tested but did not yield significant improvement compared to the relatively simple approach presented here.

The result of this model is illustrated in Fig. 1a, which depicts the temperature-dependent residence time (red line), the daily precipitation input (blue bars) and the resulting terrestrial water storage time series (blue line). The initial value of the storage ($TWS(t)$ at $t = 0$) is computed from the analytical solution for the equilibrium state of Eq. (1) given a mean precipitation input $\bar{P}$:

$$\text{TWS}(0) = \bar{P} \cdot -\log\left(\text{mean}\left(e^{-\frac{1}{\tau(t)}}\right)\right)^{-1} \tag{5}$$

Using this solution (Eq. 5) requires the assumption that the storage is close to equilibrium at the start of the reconstruction but avoids the loss of six years for model spin-up as was done in previous work (Humphrey et al., 2017). Still, we note that reconstructed TWS anomalies at the very beginning of the time series (typically the first year) should be interpreted with care.



### 2.3.2 Model calibration

The daily water storage time series (Eq. 1) is averaged to monthly temporal resolution in order to make it comparable with the monthly GRACE time series. Calibration is conducted at monthly scale against de-seasonalized and de-trended GRACE TWS observations (Fig 1b), such that:

$$\text{anom}\big(GRACE(t)\big) = \beta \cdot \text{anom}\big(TWS(t)\big) + \varepsilon \tag{6}$$

where $\beta$ is a calibrated scaling factor, $\varepsilon$ corresponds to an error term and $\text{anom}()$ is an operator indicating that the seasonal cycle and the linear trend are removed as mentioned above. The trends are removed

during model calibration because many trends in GRACE are caused by anthropogenic activities (Humphrey, 2017;Rodell et al., 2018), which our climate-driven model cannot explain by definition. The three model parameters ($a$, $b$: Eq. 2 and $\beta$: Eq. 6) are calibrated at each mascon using a Markov Chain Monte Carlo (MCMC) procedure minimizing the sum of squares of the residuals between the predicted and observed monthly TWS anomalies (Haario et al., 2006;Humphrey et al., 2017). The MCMC

procedure provides distributions of equally acceptable parameter sets which are later used in the generation of ensemble members (section 2.4).

## 2.4 Generation of ensemble members at monthly resolution

### 2.4.1 Rationale for the generation of model ensembles

The empirical residuals ($\varepsilon$) in Eq. (6) correspond to the difference between observed and predicted water

storage anomalies. They include measurement errors from GRACE, structural model errors and errors introduced by the imperfect meteorological forcing. In this section, we aim to quantify and communicate the magnitude of these errors to end users in a practical way. A classical approach is to provide the standard error $\sigma_\varepsilon$ for every mascon $m_{1,\dots,i,\dots,n}$ (Fig. 2a):

$$\sigma_\varepsilon(m_i) = \sqrt{\text{Variance}[\varepsilon(m_i)]} \tag{7}$$



Because it can be shown in our case that the residuals are normally distributed (Fig. 2b), it is relatively safe to use the standard error to estimate the predictive uncertainty (and any confidence interval) over a given mascon. However, in many applications, predictions from individual mascons need to be aggregated, for instance to compute basin-scale averages or global means. In this case, obtaining an error

estimate for the aggregated value is not trivial because the spatial covariance of the errors needs to be taken into account during the error propagation (Bevington and Robinson, 2003). Because errors are spatially and temporally correlated, any averaging operation (in the time or space domain) potentially requires that error covariance is taken into account.

To provide a practical solution to this problem, we generate ensemble members which incorporate the

spatial and temporal covariance structure of the residuals. These ensembles can be easily averaged over any larger area and once averaged, they provide a predictive spread that is representative of the aggregated error. In order to generate these ensembles, we present hereafter a spatial autoregressive (SAR) noise model (Cressie and Wikle, 2011) which aims at reproducing the spatial and temporal autocorrelation structure found in the empirical residuals ($\varepsilon$). The SAR model is used to generate random realizations of

these residuals (hereafter noted $\hat{\varepsilon}$) which have a spatial and temporal autocorrelation structure that is comparable to that of the empirical residuals ($\varepsilon$). De-seasonalized ensemble members ($GRACE_{REC}$) are obtained by combining the monthly water storage predictions (from Eq. 6) with the randomly generated residuals $\hat{\varepsilon}$.

$$\mathrm{GRACE_{REC}} = \beta \cdot deseas\big(\mathrm{TWS(t)}\big) + \hat{\varepsilon} \qquad (8)$$

### 2.4.2 Generation of random residuals

In the SAR model (Cressie and Wikle, 2011), residuals ($\hat{\varepsilon}_t$) at a given time step are represented as the sum of: 1) the product of the antecedent residual ($\hat{\varepsilon}_{t-1}$) with a local (mascon-specific) autoregressive

parameter ($\varphi$) and 2) spatially auto-correlated innovations ($\eta$) that are randomly generated from a multivariate Gaussian with zero mean and covariance matrix $\mathbf{Q_n}$:



$$\begin{bmatrix} \hat{\varepsilon}_t(m_1) \\ \vdots \\ \hat{\varepsilon}_t(m_n) \end{bmatrix} = \begin{bmatrix} \varphi(m_1) \cdot \hat{\varepsilon}_{t-1}(m_1) \\ \vdots \\ \varphi(m_n) \cdot \hat{\varepsilon}_{t-1}(m_n) \end{bmatrix} + \begin{bmatrix} \eta(m_1) \\ \vdots \\ \eta(m_n) \end{bmatrix} \tag{9}$$

where $m_{1,\dots,n}$ corresponds to the mascon index and squared brackets indicate a $n \times 1$ vector. An equivalent vector notation yields:

$$\hat{\boldsymbol{\varepsilon}}_t = \boldsymbol{\varphi} \circ \hat{\boldsymbol{\varepsilon}}_{t-1} + \boldsymbol{\eta}, \qquad \boldsymbol{\eta} \sim iid\ Gau(0, \mathbf{Q}_n) \tag{10}$$

where $\hat{\boldsymbol{\varepsilon}}_t$, $\hat{\boldsymbol{\varepsilon}}_{t-1}$, $\boldsymbol{\varphi}$ and $\boldsymbol{\eta}$ are $n \times 1$ vectors, $\mathbf{Q}_n$ is a $n \times n$ spatial covariance matrix and $\circ$ denotes the Hadamard product (i.e. pair-wise multiplication).

10   The local autoregressive parameters $\varphi(m_1, \dots, m_n)$ are estimated at each mascon from the lag-1 temporal autocorrelation of the empirical residuals ($\varepsilon$) ($\varphi$ illustrated in Fig 2c) (Wilks, 2011). To estimate the spatial covariance matrix of the innovations ($\mathbf{Q}_n$), we follow the following procedure. First, an isotropic exponential decay autocorrelation function (Eq. 11) is fitted at each individual mascon (Fig 3a, b) to represent the spatial autocorrelation (AC) of the empirical residuals, such that:

$$AC(d) = e^{-\frac{d}{k}} \tag{11}$$

where $d$ is the distance and $k$ is the parameter to fit. Locations with high (low) values of $k$ (Fig 3c) indicate regions where the residuals have a strong (weak) spatial autocorrelation. The calibrated AC
20   functions are then used to construct the spatial autocorrelation matrix $\mathbf{P}_n$ which approximates the structure of the spatial autocorrelation matrix of the empirical residuals. From this, the covariance matrix for the innovations is obtained by definition as:

$$\mathbf{Q}_n = \text{diag}(\boldsymbol{\sigma}_\eta)\ \mathbf{P}_n\ \text{diag}(\boldsymbol{\sigma}_\eta) \tag{12}$$





where $\boldsymbol{\sigma_\eta}$ is a $n \times 1$ vector containing the standard deviation of the innovations at each mascon estimated from (Cressie and Wikle, 2011):

$$\boldsymbol{\sigma_\eta} = \boldsymbol{\sigma_\varepsilon} \circ \sqrt{1 - \boldsymbol{\varphi}^2} \qquad (13)$$

where $\boldsymbol{\sigma_\varepsilon}$ is the empirical standard error of each mascon (Eq. 7, Fig 2a). The multiplication with $\sqrt{1 - \boldsymbol{\varphi}^2}$ scales the empirical standard error under the assumption of an autoregressive process of order 1 (Cressie and Wikle, 2011). This accounts for the fact that the variance of an autoregressive process is larger than that of the driving white noise process. In the special case where the first residual in Eq. (10) ($\hat{\boldsymbol{\varepsilon}}_t$ at $t = 1$)

10  is generated and $\hat{\boldsymbol{\varepsilon}}_{t-1}$ does not exist yet, the multiplication with $\sqrt{1 - \boldsymbol{\varphi}^2}$ is not necessary and the following formulations are used instead of Eq. (10) and (12):

$$\hat{\boldsymbol{\varepsilon}}_1 = \boldsymbol{\eta}, \qquad \boldsymbol{\eta} \sim iid\ Gau(0, \mathbf{Q'}_n) \qquad (14)$$

$$\mathbf{Q'}_n = \text{diag}(\boldsymbol{\sigma_\varepsilon}) \cdot \mathbf{P}_n \cdot \text{diag}(\boldsymbol{\sigma_\varepsilon}) \qquad (15)$$

To summarize, a first residual is generated with Eq. (14) and subsequent residuals are generated from Eq. (10).

As mentioned in section 2.3, the Markov Chain Monte Carlo (MCMC) procedure for model parameter
20  estimation additionally provides a distribution of equally acceptable model parameters ($a$, $b$ and $\beta$). Each parameter set provides one ensemble member for which the entire procedure described here is repeated. Thus, ensemble members combine 1) a structural model uncertainty arising from the distribution of calibrated model parameters and 2) an estimate of the predictive uncertainty. Here, we provide one hundred randomly sampled ensemble members. This number was chosen as a compromise between the
25  size of the final dataset and the minimum number of ensemble members required to derive a reasonable estimate of the 90% confidence interval.





### 2.4.3 Evaluation of ensemble members

The result of the above-described procedure is briefly illustrated and evaluated in Fig. 4. For illustration, Fig. 4a shows the empirical residuals ($\varepsilon$) for the month of April 2002 and Fig. 4b shows one instance of the randomly generated residuals ($\hat{\varepsilon}$). As expected, both the empirical and the randomly generated

residuals exhibit spatial autocorrelation. The generated residuals also have approximately the same variance (Fig. 4c) and lag-1 temporal autocorrelation (Fig. 4d) as that of the empirical residuals. The confidence intervals derived at a regional or basin-scale level reliably cover the actual GRACE-based regional average which was the initial motivation for the presented approach (illustrated for the Mississippi basin in Fig. 4e). We evaluate the overall *reliability* of the ensemble hindcast for regional

averages over 90 large (>500'000 km$^2$) river basins using a rank histogram (or Talagrand diagram) (Fig. 4f). In the ideal case (*perfect reliability*), the observed TWS ranks lower than the $P^{th}$ percentile of the reconstruction only $P$ percent of the time (for instance, GRACE observations should be lower than the 5$^{th}$ percentile of the reconstruction only 5% of the time). According to this first order metric (see e.g. Hamill, 2001 for a discussion), we conclude that regional averages of the ensemble members provide *reliable*

forecasts (Fig. 4f), with only a minor tendency to miss extreme positive TWS anomalies.

The presented method represents one amongst many possible approaches to the generation of ensemble members. This method has the advantage of reflecting the uncertainty of the reconstruction (compared to GRACE measurements) and mimics the empirical spatiotemporal auto-correlation structure of the errors while only requiring a minimal degree of model complexity and parameterization. We note that while the

SAR model also represents errors coming from the GRACE solution itself, it does not include any anisotropic error structure (e.g. due to striping) due to the isotropic nature of Eq. (11). The uncertainty related to the choice of the input precipitation or training GRACE dataset can be explored independently by comparing the six different versions of GRACE-REC (see Table 3).

Finally, we note that our modelling approach could in principle be evaluated with a cross-validation

experiment, using only a subset of the data to calibrate the model parameters and then evaluate the performance against the other "unused" data (as done in Humphrey et al., 2017). However, this would go beyond the scope and objective of this paper which is to document the generation of the GRACE-REC product. We prefer to evaluate the ability of the final product to extrapolate beyond the model calibration



period in later sections by comparing the model predictions with fully independent datasets (Sections 4.3 to 4.5).

## 3 Product description

### 3.1 Definition of GRACE-REC TWS datasets

The GRACE-REC data provide de-seasonalized terrestrial water storage (TWS) anomalies in units of millimetres ($kg/m^2$) (Eq. 8). Thus, GRACE-REC does not include a reconstructed seasonal TWS cycle. Because some applications also require the seasonal signals, we provide the GRACE-based TWS seasonal cycle (Humphrey et al., 2017) which can directly be added to the GRACE-REC TWS anomalies if needed. As a caveat, note that this GRACE-based TWS seasonal cycle is kept constant over time, which might

potentially be unrealistic.

### 3.2 Monthly products with ensemble members

Using two different training GRACE datasets (Table 1) and three different precipitation forcing datasets (Table 2), we produce a total of six different GRACE-REC datasets with 100 ensemble members each. For convenience, we also provide smaller summary files which only contain the ensemble mean and 90%

confidence interval.

### 3.3 Daily products

For the daily TWS reconstructions, we only provide the ensemble mean of each GRACE-REC product in order to limit the data size. This ensemble mean is based on ensemble members which sample the parameter uncertainty only (Section 2.3.2). The format is identical to that of the monthly data (Table 3).

### 3.4 Global averages

For global-scale applications, we provide global averages of the TWS time series. Global averages are weighted by mascon area and include all land mascons with or without Greenland and Antarctica (both options are available). This format is especially suited for sea level and global water budget studies and units are gigatons of water. To convert gigatons back to millimetres of global land water, total land area



values of 148'940'000 km$^2$ and 132'773'914 km$^2$ can be used for each option respectively. The evaluation of global means in Sections 4.1.2 and 4.3 can guide the choice between the different versions of GRACE-REC.

### 3.5 Important limitations and caveats

Although linear trends are removed during model calibration (Eq. 6), potential TWS trends caused by long-term changes in precipitation are not removed from the final dataset (Eq. 8). By definition, any trend found in the reconstructed TWS products is caused by a trend in the underlying precipitation forcing (since the time-varying residence time is using de-trended temperature and there is no limit to storage capacity). With this in mind, it should be clear that there will be differences between the trends found in
GRACE and the trends found in the reconstruction. Such discrepancies are expected because the reconstruction does not represent several sources of long-term changes in TWS, including for instance, land ice melt, anthropogenic water depletion or long-term changes in evaporative demand. As illustrated in Humphrey et al. (2017), the reconstruction can be used to remove the precipitation-driven variability from the original GRACE time series in order to better isolate and quantify these other sources of long-
term changes. However, users interested in computing long-term TWS trends from this dataset should always proceed with caution as the dataset was not evaluated for trends. For regional analyses, we recommend to use the model ensembles to obtain a range of possible trends and thus better assess the uncertainty.

  More generally, we highlight that the quality of the reconstruction is strongly dependent on the quality of
the input precipitation forcing and on the adequateness of an exponential decay model for representing water storage behaviour. For instance, routing of water through the river system is not represented and might be important over certain regions. Section 4.1 provides global maps of model performance that can guide regional applications.



## 4 Product evaluation

### 4.1 Comparison with de-seasonalized monthly GRACE

#### 4.1.1 Mascon scale

In this section, the ensemble mean of GRACE-REC is compared against GRACE observations. Note that
this does not constitute an independent evaluation because GRACE-REC is calibrated with GRACE data
(comparisons with independent sources are provided in sections 4.3 to 4.5). We evaluate model
performance with the Pearson correlation coefficient (Fig. 5) and the Nash-Sutcliffe Efficiency (Fig. 6).
Model performance is highest especially in regions with dense meteorological observing systems (e.g.
Europe, Western Russia, North America, India, Australia) where we expect precipitation datasets to have
the highest accuracy. Over South America and Central Africa, the performance of reanalysis-based
precipitation (ERA-Interim based products, Fig. 5e-f and 6e-f) is inferior to that of multi-source
precipitation datasets such as GSWP3 and MSWEP (note that GSWP3 is bias-corrected with GPCP,
which is using both station and satellite data). Interestingly, there is no clear difference in performance
when GRACE-REC is calibrated with the 3° JPL Mascons (left column) or the 1° GSFC Mascons (right
column). We conclude that in terms of model performance, the choice of the GRACE product used to
calibrate GRACE-REC is of secondary importance compared to the accuracy of the input precipitation
datasets.

We compare these performance metrics with the scores obtained by hydrological models and land surface
models of the Water Resources Reanalysis version 2 (WRR2) (Schellekens et al., 2017;Dutra et al., 2017),
which were also forced with MSWEP precipitation. Compared to the simple modelling approach used in
GRACE-REC, WRR2 models are forced with additional meteorological information (such as radiation
and humidity), were calibrated using various data streams, sometimes including GRACE observations
(Dutra et al., 2017;Decharme et al., 2011;Decharme et al., 2012;Vergnes et al., 2014;Decharme et al.,
2016;Krinner et al., 2005;de Rosnay et al., 2002;Van Der Knijff et al., 2010;Döll et al., 2009;Sutanudjaja
et al., 2011, 2014;van Beek and Bierkens, 2008;van Beek et al., 2011;Wada et al., 2011;Wada et al.,
2014;van Dijk et al., 2013;van Dijk et al., 2014), and are potentially able to resolve more complex
processes that are relevant for TWS, such as snow dynamics, the effect of vegetation phenology on



evapotranspiration, and runoff routing through the river system. We calculate TWS in WRR2 models by summing over all simulated water reservoirs (this includes soil moisture, snow, groundwater and surface waters whenever these are represented in the models). It is important to underline that unlike WRR2 models, GRACE-REC is directly calibrated to reproduce GRACE observations. Therefore, GRACE-REC

should be interpreted here as a benchmark, indicative of the performance that is at least achievable for a given precipitation dataset. In terms of Nash-Sutcliffe efficiency, GRACE-REC often obtains better scores than the WRR2 models (Fig 7a). This is because the reconstruction better fits the local amplitude and variance of the observed TWS signal, as already diagnosed in previous work (Humphrey et al., 2017). We note that the reconstructions driven with MSWEP precipitation are systematically superior to those

driven with the other two precipitation datasets (note WRR2 is driven with MSWEP as well).

### 4.1.2 Global scale

Global averages of all GRACE-REC products are illustrated in Fig. 8a. Differences caused by different precipitation forcing datasets are much greater than the differences related to different GRACE training datasets. This is particularly true for long-term (> 20 years) trends as we find that, over the overlapping

period 1979-2014, the two MSWEP-based products both produce a positive climate-driven TWS trend while GSWP3-based products yield a negative TWS trend. We conclude that long-term trends in GRACE-REC should be interpreted with the awareness that reconstructed TWS trends strongly depend on the trends initially present in the driving precipitation data. Note that trends in GRACE-REC cannot be directly evaluated against the trends from GRACE itself. This is because GRACE-REC only represents

precipitation-driven effects, whereas GRACE observations also include effects of groundwater depletion, dams and glacier melt (Reager et al., 2016;Felfelani et al., 2017;Rodell et al., 2018) as well as potential effects of climate change on evaporative demand.

Comparisons with the de-trended GRACE global average are shown in Fig. 8b-c. We find that all GRACE-REC products compare well against actual global mean GRACE, and this without applying any

global constraint to the locally calibrated statistical model. Correlations between global means of GRACE-REC and global means of GRACE are in general larger than 0.75 (Fig 9a) (evaluated over the common period 2003-2014). Compared to global means from the WRR2 models, GRACE-REC is on



average better correlated (Fig. 9a) to the observed GRACE global mean and has a lower root mean square error (Fig. 9b), regardless of the GRACE dataset used for evaluation.

### 4.2 Comparison with de-seasonalized daily GRACE

We compare the daily GRACE-REC products with a Kalman smoothed daily GRACE solution named 5 ITSG2018 (Kurtenbach et al., 2012;Mayer-Gürr et al., 2016). While this daily GRACE solution contains significant information on the sub-monthly variability of TWS, the increased temporal resolution is at the cost of spatial resolution, which is in the order of 500km for this particular product. As illustrated in Figure 10a, there can be a good agreement between GRACE-REC and ITSG2018 for submonthly variability when daily averages are computed over large regions (here the Mississippi basin). Figure 10b- 10 c provides a summary of the agreement between GRACE-REC and ITSG2018 at daily scale, as well as a comparison with the performance of WRR2 models. Due to the coarse resolution of the ITSG2018 product, the comparison (Fig. 10b-c) is conducted at a spatial resolution of 5°. We find that, even though the performance of all products is lower than at monthly resolution, the GRACE-REC products agree on average as well or better with ITSG2018 than most models of the WRR2 ensemble.

### 15 4.3 Comparison with the de-seasonalized and de-trended sea level budget

Together with changes in ocean heat content, changes in the amount of water stored on land are responsible for a large fraction of the year-to-year variability in global mean sea level (Boening et al., 2012;Cazenave et al., 2014;Cazenave, 2018). Because changes in land water storage result in opposite changes in ocean mass, the sea level budget provides an independent mean of evaluating various estimates 20 of global mean TWS variability. Here we assess the ability of terrestrial water storage products (GRACE, GRACE-REC, and the WRR2 models) to close the sea level budget at the inter-annual time scale. We use de-seasonalized and de-trended global mean sea level (GMSL) from satellite altimetry (Beckley et al., 2017) and steric height estimates ($GMSL_{steric}$) based on observations of Argo floats (Roemmich and Gilson, 2009;Llovel et al., 2014). From the sea level budget, we obtain an estimate of inter-annual changes 25 in ocean mass (Eq. 16, black line in Fig. 11a) which we compare against global mean TWS estimates. We use this budget-based ocean mass to provide an independent evaluation of all TWS products (i.e. not



based on any GRACE data), although GRACE-based ocean mass is obviously also available since 2002. Greenland and Antarctica are excluded from the TWS averages to enable a consistent comparison among all products (hydrological models typically do not represent these regions).

$$GMSL_{ocean\ mass} = GMSL - GMSL_{steric} \tag{16}$$

We find that, although all considered products are significantly correlated with the budget-based ocean mass ($GMSL_{ocean\ mass}$), GRACE and GRACE-REC estimates are clearly better correlated and yield a lower root mean square error (Fig. 11b-c). Surprisingly, GRACE-REC products also yield better results

than the two original GRACE datasets (JPL and GSFC). We hypothesize that this might occur because the global mean GRACE TWS is more susceptible to non-compensating continental-scale errors (e.g. caused by errors in low degree spherical harmonics) compared to climate-driven reconstructions which yield smoother global averages (as seen in Fig. 8b,c).

### 4.4 Comparison with de-seasonalized basin-scale water balance

Over moderately large river basins (>100'000km²), TWS changes can be estimated by combining streamflow measurements with moisture fluxes from an observation-assimilating atmospheric reanalysis system (Oki et al., 1995;Seneviratne et al., 2004). This approach provides relatively independent estimates of TWS changes over large basins which has been used to evaluate distributed hydrological models and land surface models. Here, we aim to use such estimates to evaluate the quality of the reconstruction also

during the period where no GRACE data is available (i.e. prior to 2002).
We evaluate TWS products using a recently updated basin-scale water balance dataset (BSWB) (Hirschi and Seneviratne, 2017) which covers 341 catchments and is based on ERA-Interim reanalysis data and runoff observations from the Global Runoff Data Centre (GRDC). As a caveat, we note that BSWB should not be viewed as entirely independent from WRR2 models neither as a ground truth. This is because

moisture fluxes from ERA-Interim are not only influenced by the assimilated atmospheric profile information but are also dependent on the underlying land surface model (TESSEL), which is similar to WRR2 models in many aspects. All WRR2 models also used ERA-Interim as forcing data for all





meteorological variables except for precipitation. Similarly, the JPL-MSWEP and GSFC-MSWEP reconstructions use ERA-Interim air temperature, while JPL-ERAI and GSFC-ERAI reconstructions use both ERA-Interim precipitation and air temperature.

As illustrated in Fig 12a for the Ob basin, we find that the reconstructed TWS compares relatively well

with BSWB estimates. Overall, all TWS products considered here (including the GRACE data itself) seem to compare equally well with BSWB (Fig 12b-c). We note that GRACE GSFC data and GRACE-REC products calibrated on GSFC seem to compare slightly better with BSWB than the JPL-based products. This might be because of the higher spatial resolution of the GSFC mascons (1° instead of 3° for JPL) which might enable a better separation between mass changes located inside or outside the river

basin boundaries.

## 4.5 Comparison with annual streamflow measurements

In this section, we compare reconstructed TWS against streamflow observations over the period 1901 to 2010. Streamflow and TWS of course represent different variables with different units, however, we expect that their temporal dynamics will correlate at the yearly scale, as illustrated for the river Thames

in Fig 13a-b. Because observed streamflow is one of the few water cycle variables available prior to 1980, it provides an independent and useful means of evaluating the century-long reconstruction. We use streamflow observations collected by the Global Streamflow Indices and Metadata Archive (GSIM) (Do et al., 2018;Gudmundsson et al., 2018). From the 30'959 available stations, we keep stations with basin size smaller than 10'000 km$^2$ and with at least 10 years of available data (discarding any year where less

than 50% of the daily values were available to compute the yearly mean), leaving 12'496 stations for analysis.

We find that TWS anomalies from both WRR2 models and GRACE-REC compare well with yearly streamflow variability over the period 1980-2010 (Fig. 13c). Reconstructions based on the GSFC products tend to perform slightly better, again likely because of their higher spatial resolution (1°) compared to the

JPL-based reconstructions (3°). When evaluating the century-long reconstruction (GSWP3-driven products), we find that the correlation between yearly TWS anomalies and yearly runoff only slightly degrades for the earliest time period (1901-1940) but is otherwise relatively stable over time (Fig. 13d).





This indicates that, even though GRACE-REC was calibrated over the years 2002-2016, the model is still able to reproduce past water cycle variability and does not overfit to the period of the GRACE mission. In addition, we note that the quality of the century-long reconstruction is of course dependent on the accuracy of the GSWP3 precipitation and temperature forcing, which likely degrades towards the
beginning of the century as less observations are available.

## 5 Data availability

The presented dataset is publicly available (https://doi.org/10.6084/m9.figshare.7670849) and updates will be published when needed. The data can be freely used provided this paper is acknowledged.

## 6 Conclusions

We present a statistical reconstruction of climate-driven terrestrial water storage changes at daily and monthly resolution in six different configurations which cover three different time periods (Table 3). We evaluate the performance of this reconstruction and show that its overall accuracy is reasonable compared to other estimates of TWS variability available from global hydrological models. We also highlight the versatility and robustness of our approach by comparing our estimates with independent observations of
Earth system variables outside of the calibration period.

## 7 Author contribution

VH and LG developed the approach. VH performed the analyses, produced the dataset and wrote the manuscript with feedback from LG.





## 8 Competing interests

The authors declare that they have no conflict of interest.

## 9 Acknowledgements

This research was funded by the European Research Council DROUGHT- HEAT project (contract
617518). We thank Prof. Dr. Sonia Seneviratne for critical feedback and support of this work. We thank
Prof. Dr. Hyungjun Kim for developing the GSWP3 forcing and providing us with early access to the
data (https://doi.org/10.20783/DIAS.501). We thank Dr. Richard Wartenburger for technical support.
Model developers and data providers are also gratefully acknowledged for sharing their data: GRACE
JPL    Mascons    (https://grace.jpl.nasa.gov/data/get-data/jpl_global_mascons/),    GRACE    GSFC
(https://neptune.gsfc.nasa.gov/gngphys/index.php?section=470), MSWEP V2 (http://www.gloh2o.org/),
ERA-Interim (http://apps.ecmwf.int/datasets/data/ interim-full-daily), ITSG2018 (http://icgem.gfz-
potsdam.de/series),    NASA    Sea    Level    Change    Portal    (https://sealevel.nasa.gov/),    BSWB
(doi:10.5905/ethz-1007-82), GRDC Reference Dataset (https://www.bafg.de/GRDC), GSIM
(https://doi.org/10.1594/PANGAEA.887477),    WRR2    (http://wci.earth2observe.eu/thredds/catalog-
earth2observe-model-wrr2.html) and the Earth2Observe project (http://www.earth2observe.eu/).

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



*Table 1. GRACE datasets used for model calibration*

| GRACE product | Time period | Spatial resolution | GIA correction | Access | Citation |
|---|---|---|---|---|---|
| JPL-Mascons RL05 with CRI | April 2002 - June 2017 | 3° equal-area mascons, sampled on a 0.5° grid | (A et al., 2013) | ftp://podaac.jpl.nasa.gov/ allData/tellus/L3/mascon/ RL05/JPL/CRI/ | (Watkins et al., 2015;Wiese et al., 2016) |
| GSFC-Mascons v2.4, ICE6G | January 2003 - July 2016 | 1° equal-area mascons, sampled on a 0.5° grid | (Peltier et al., 2015) | https://neptune.gsfc.nasa.gov/ gngphys/index.php? section=456products.html | (Luthcke et al., 2013) |



*Table 2. Meteorological forcing datasets*

| Dataset | Time period | Spatial resolution used | Description | Access | Citation |
|---------|-------------|-------------------------|-------------|--------|----------|
| MSWEP v2.2 | 1979-2016 | 0.5° grid | Merged precipitation product combining multiple data sources | http://www.gloh2o.org/ | (Beck et al., 2018) |
| ERA-Interim | 1979-current | 0.5° grid | Atmospheric reanalysis with regular updates | http://apps.ecmwf.int/datasets/data/interim-full-daily/levtype=sfc/ | (Dee et al., 2011) |
| GSWP3 v1.1 | 1901-2014 | 0.5° grid | ERA 20th Century Reanalysis, downscaled to 0.5° resolution using spectral nudging and bias-corrected with GPCP and CRU | http://www.dias.nii.ac.jp/gswp3/input.html | (Kim, 2017) |



Table 3. List of the 6 GRACE-REC datasets available at monthly and daily scale

| GRACE-REC dataset | Time period | Spatial resolution | Forcing data | Training data | Unit |
|---|---|---|---|---|---|
| JPL-MSWEP | 1979-2016 | 3° equal-area (provided on a 0.5° grid) | MSWEP | GRACE JPL | mm TWS |
| JPL-GSWP3 | 1901-2014 | | GSWP3 | | |
| JPL-ERAI | 1979-current | | ERA-Interim | | |
| GSFC-MSWEP | 1979-2016 | 1° equal-area (provided on a 0.5° grid) | MSWEP | GRACE GSFC | |
| GSFC-GSWP3 | 1901-2014 | | GSWP3 | | |
| GSFC-ERAI | 1979-current | | ERA-Interim | | |



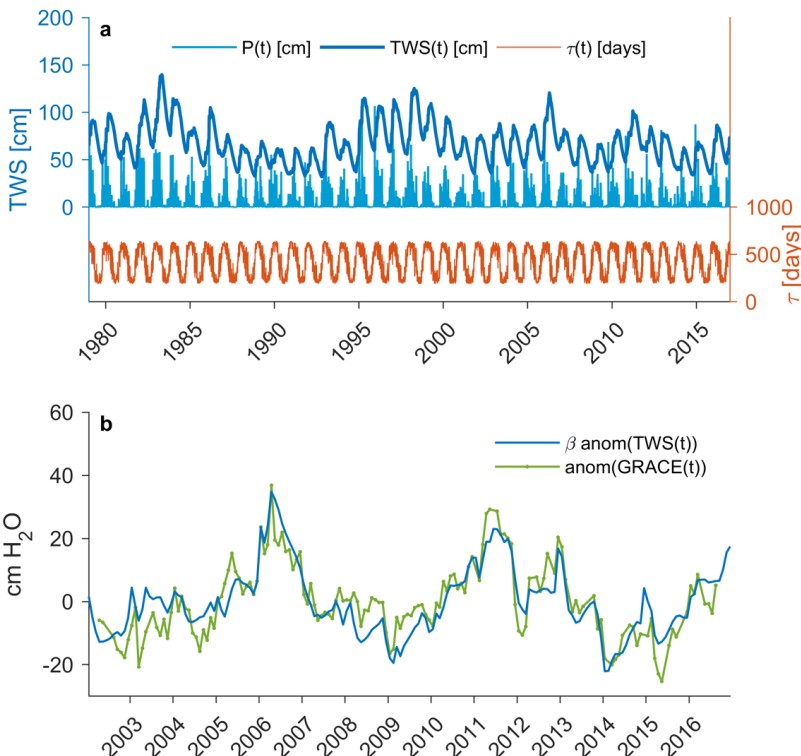

*Figure 1. Illustration of the GRACE reconstruction at one given 3° x 3° mascon (located in California). (a) Input daily precipitation time series $P(t)$, temperature-dependent residence time $\tau(t)$, and the resulting daily TWS time series $TWS(t)$. (b) Agreement between GRACE and GRACE-REC after subtracting the seasonal cycle and long-term trend (zoomed over the period 2002-2017).*



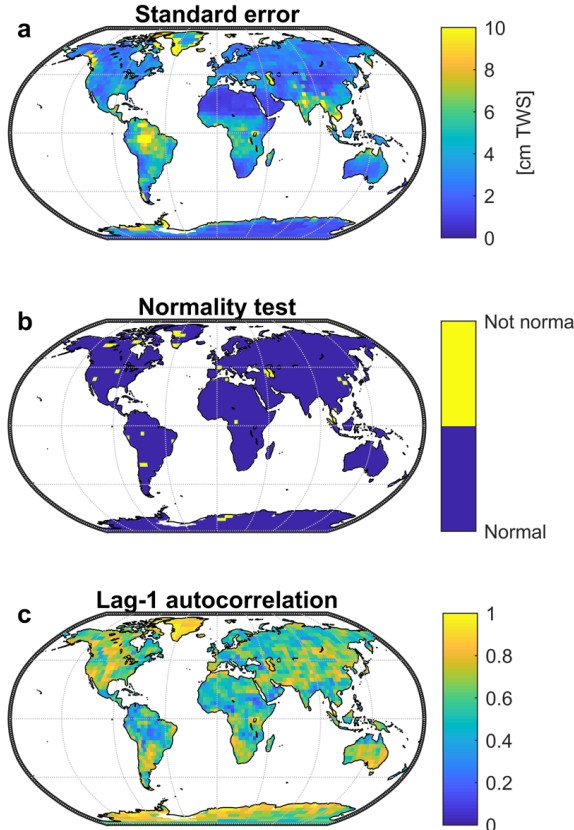

Figure 2. Characterization of the empirical model residuals for the GRACE-REC dataset based on MSWEP precipitation and ERA-Interim air temperature, calibrated with the JPL mascons. (a) Standard model error, (b) Result of a Kolmogorov-Smirnov test for normality on the model errors ($p < 0.05$), (c) lag-1 serial autocorrelation of the model errors.




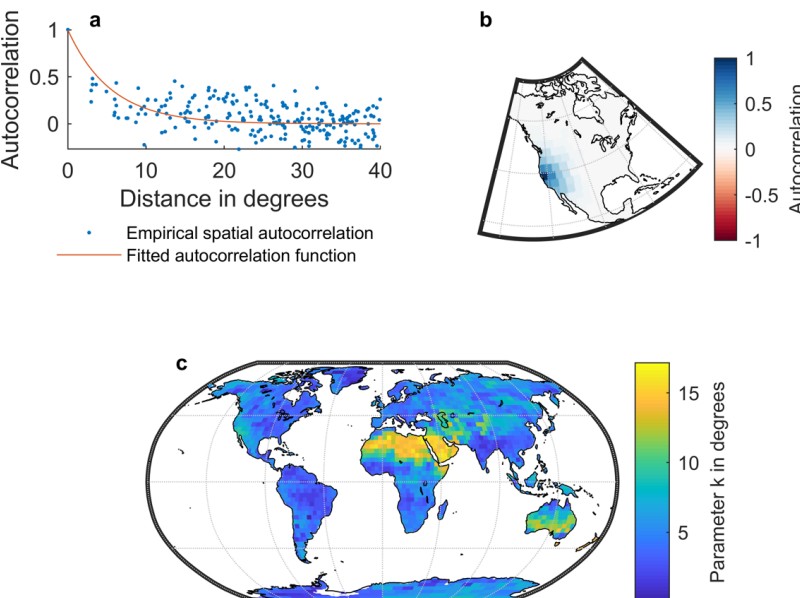

Figure 3. Illustration of the spatial autocorrelation of the empirical model residuals and their representation in the SAR model (also for the GRACE-REC product based on MSWEP and calibrated with JPL Mascons). (a) Empirical and fitted spatial autocorrelation functions for the model residuals at a given 3° x 3° mascon in California. (b) Fitted spatial autocorrelation at that mascon. (c) Fitted parameter k (Eq. 11), which conditions the steepness of the autocorrelation function (high values = high autocorrelation length of the residuals).





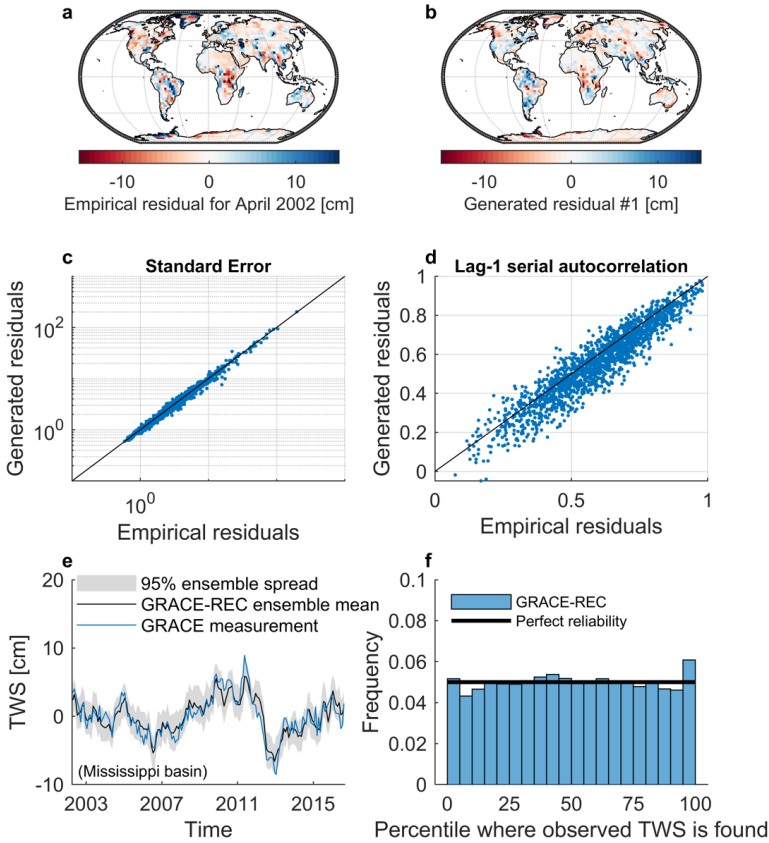

Figure 4. Output of the SAR model for the generation of random noise realisations that have a spatio-temporal structure similar to that of the empirical model residuals. (a) Empirical model residual at a given time step. (b) Residual randomly generated by the SAR model. (c) Agreement between the standard deviation of the empirical versus generated residuals. (d) Agreement between the lag-1 autocorrelation of the empirical versus generated residuals. (e) Illustration of the resulting ensemble spread for a basin-scale average. (f) Rank histogram using 5% bins, combining the data for 90 large (>500'000 km$^2$) basins (from 2003 to 2014), used to evaluate the reliability of ensemble forecasts.





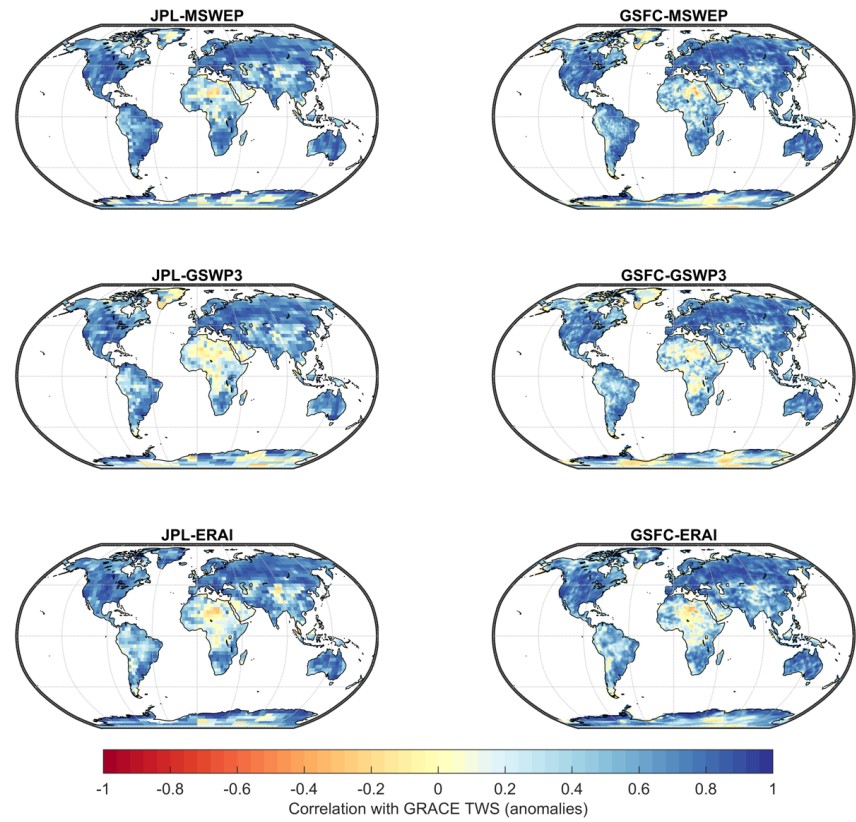

Figure 5. Correlation (of de-seasonalized, de-trended anomalies) between GRACE-REC and GRACE JPL Mascons (left column), or GRACE GSFC Mascons (right column). Three different precipitation forcing datasets are tested: MSWEP (top row), GSWP3 (middle row), and ERA-Interim (bottom row). Values closer to one correspond to a higher model performance.



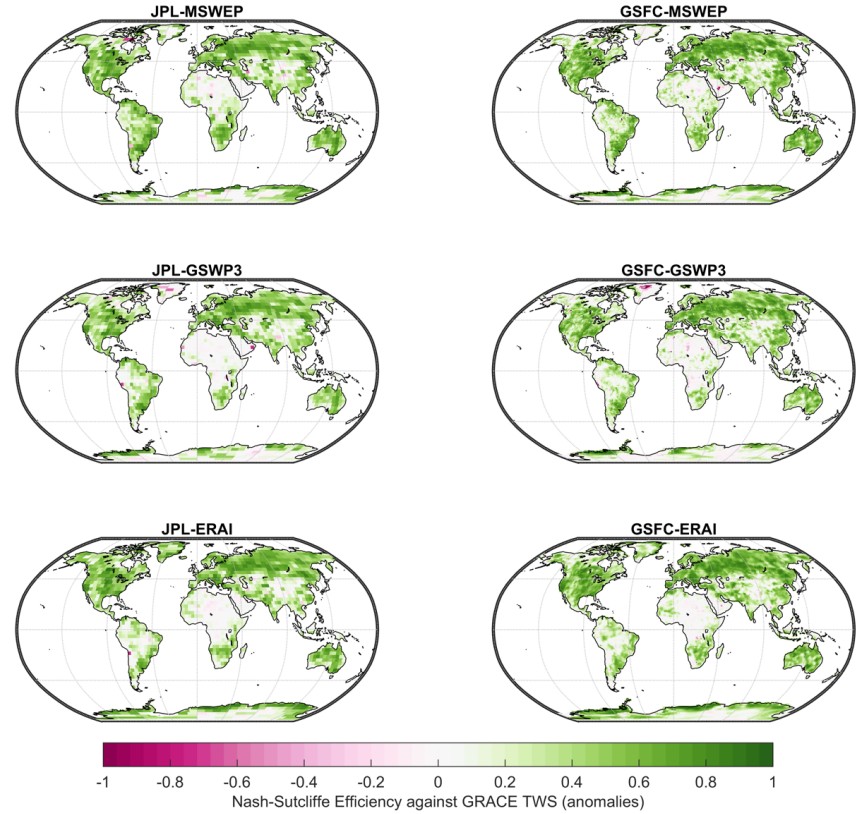

Figure 6. Nash-Sutcliffe Efficiency (of de-seasonalized, de-trended anomalies) between GRACE-REC and GRACE JPL Mascons (left column), or GRACE GSFC Mascons (right column). Three different precipitation forcing datasets are tested: MSWEP (top row), GSWP3 (middle row), and ERA-Interim (bottom row). Values closer to one correspond to a higher model performance.



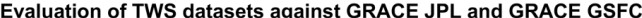

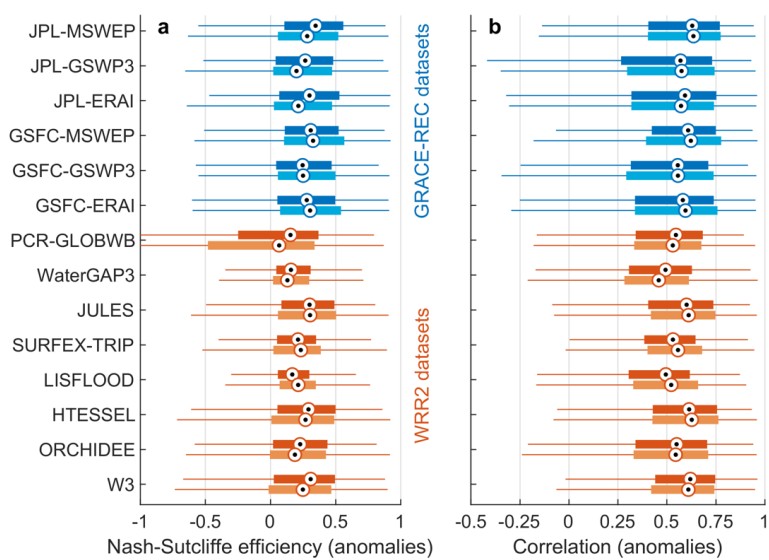

Figure 7. Global summary of the performance metrics shown in Figures 5 and 6 for GRACE-REC datasets
5  (blue), and comparison with the performance of global hydrological models participating in the
Earth2Observe Water Resources Reanalysis version 2 (WRR2) (orange). Dark colors indicate the
performance obtained when comparing against 3° x 3° JPL Mascons, and against 1° x 1° GSFC Mascons
for light colors. Note: WRR2 models are driven with MSWEP precipitation and all model outputs are
aggregated to the resolution of the corresponding GRACE dataset.





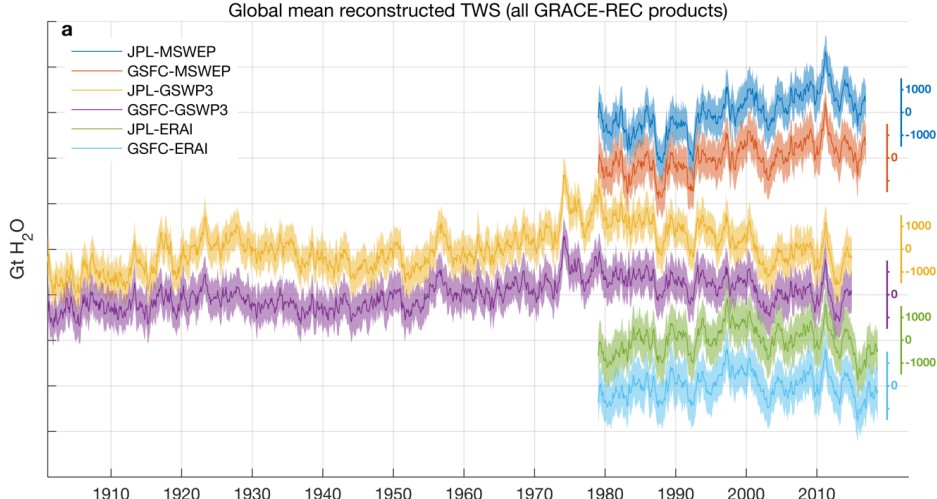

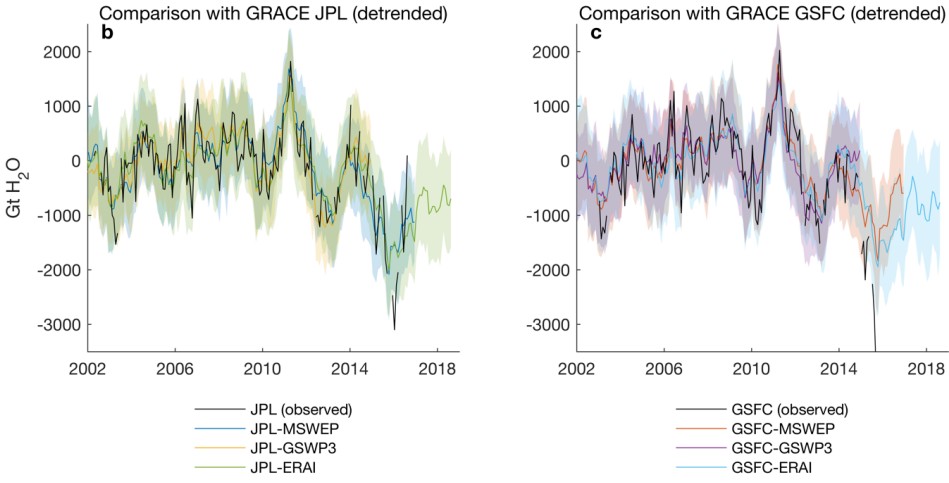

Figure 8. (a) Global average of TWS anomalies for the 6 GRACE-REC datasets (excluding Greenland and Antarctica) with an artificial vertical offset added for better visual comparison. (b) Comparison of the 3 GRACE-REC datasets calibrated with GRACE JPL against GRACE JPL (de-trended anomalies).
5    (c) Same as (b) but for GRACE GSFC.





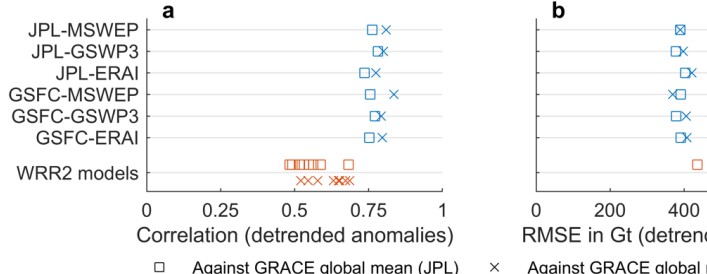

Figure 9. Agreement of the global average of different TWS model estimates (from GRACE-REC (blue) and WRR2 models (orange)) with the observed TWS anomalies from JPL (squares) and GSFC (crosses) solutions.



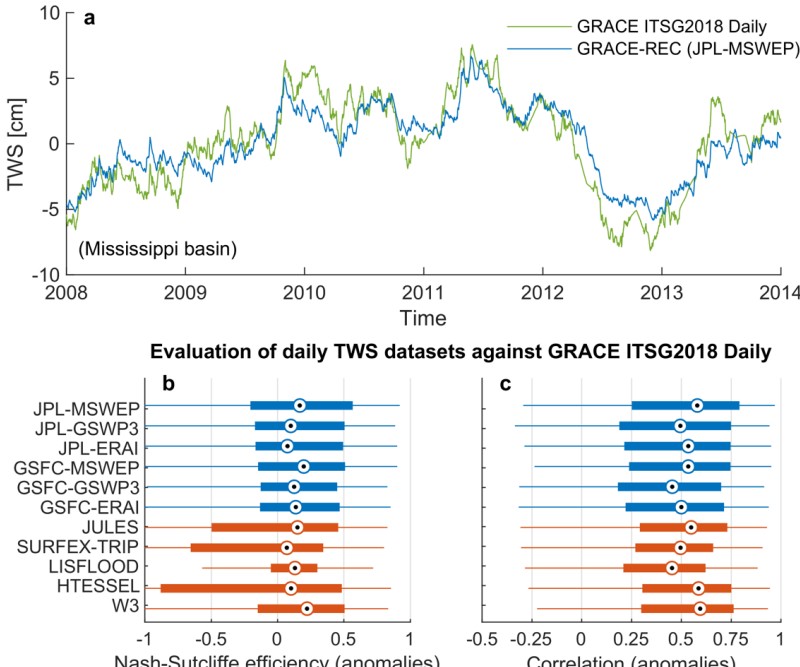

Figure 10. (a) Comparison between the GRACE-REC daily TWS reconstruction (JPL-MSWEP dataset) and the daily GRACE ITSG2018 solution for the Mississippi basin (focused over the period 2008-2014 to improve readability of the high-frequency fluctuations). (b-c) Summary of the performance metrics of the daily TWS datasets when compared with ITSG2018 at a spatial resolution of 5°. Note that some WRR2 models are not included because not all water storage variables were available to us at daily frequency.





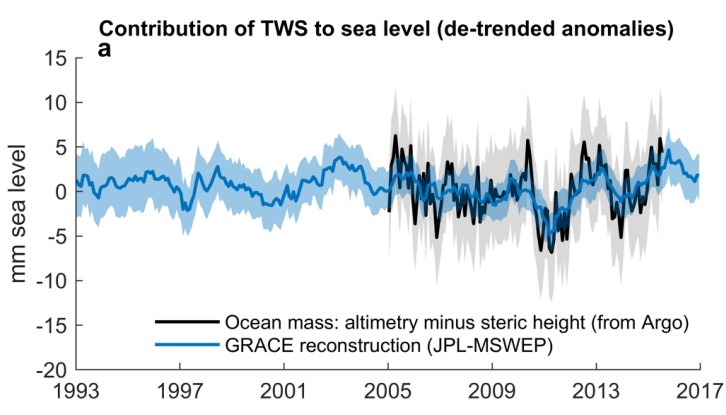

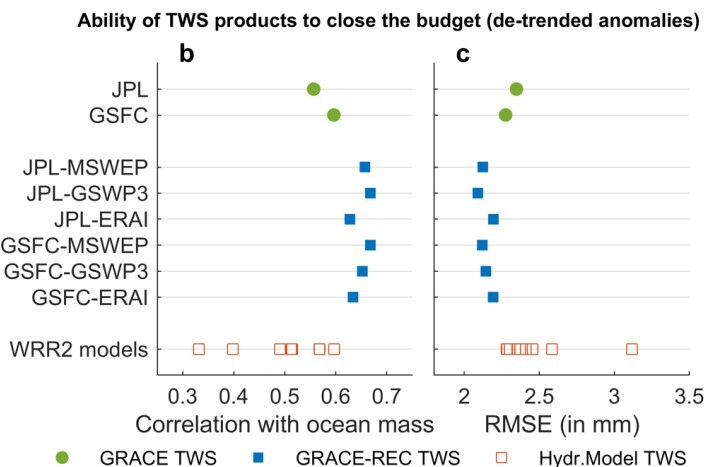

Figure 11. (a) Comparison of the global mean TWS reconstructed by GRACE-REC (converted to
5  equivalent mm sea level) against the ocean mass derived from the sea level budget. (b-c) Evaluation of
the ability of various TWS datasets to close the sea level budget (GRACE estimates in green, GRACE-
REC datasets in blue, and WRR2 models in orange).





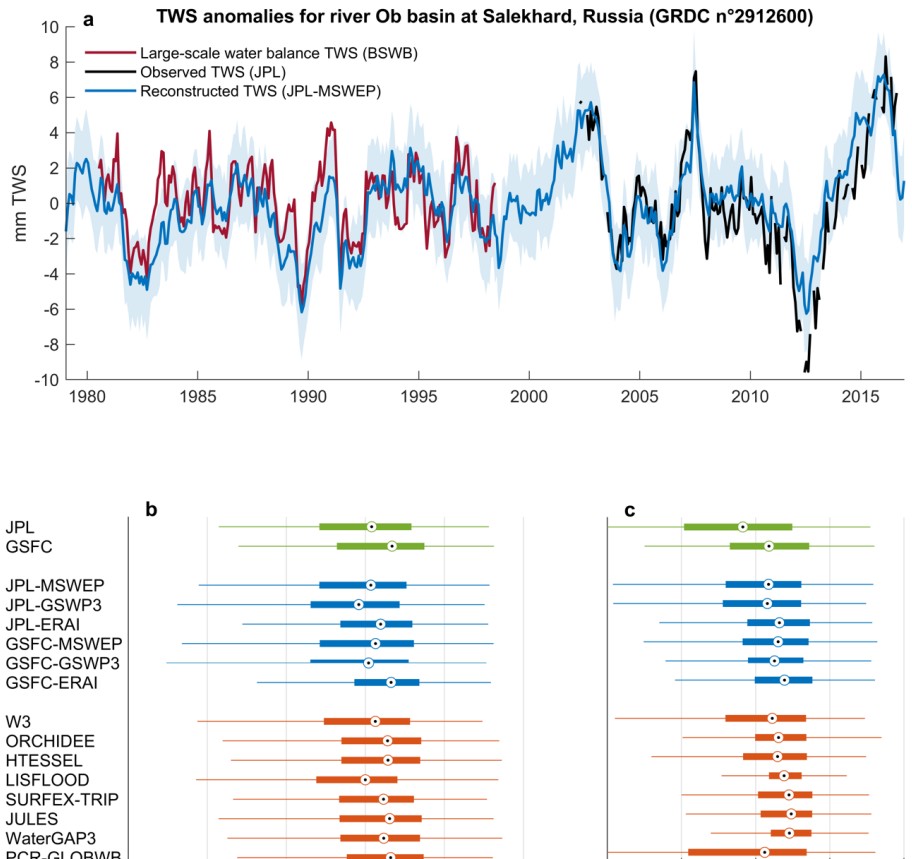

Figure 12. (a) Comparison between TWS anomalies derived from atmospheric basin-scale water
balance (BSWB), GRACE observations (JPL) and the GRACE reconstruction (JPL-MSWEP dataset).
5    (b-c) Overall evaluation of the agreement between various TWS products and BSWB estimates
(performance metrics of 341 large basins).


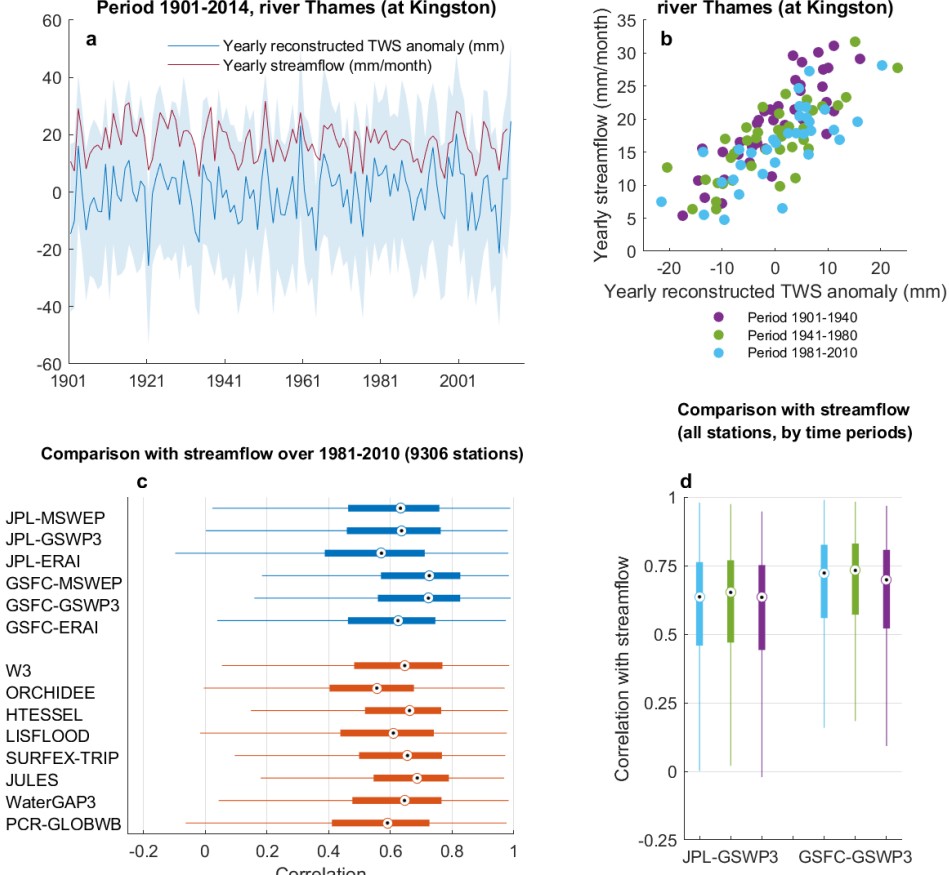

Figure 13. (a) Comparison between century-long measurements of streamflow and the TWS anomalies reconstructed at this location (GSFC-GSWP3 dataset). (b) Scatter plot of the data in (a), by time period. (c) Evaluation of the performance of GRACE-REC and WRR2 models when compared with yearly streamflow anomalies. (d) Evaluation of the performance of the JPL-GSWP3 and GSFC-GSWP3 products when compared with yearly streamflow anomalies, by time period.