# Peer review of "GRACE-REC: a reconstruction of climate-driven water storage changes over the last century"

_Earth System Science Data, 2019_

## Referee Comment (RC1) · Anonymous Referee #1 · 20 Mar 2019

General Comments: This manuscript describes a new data product that represents reconstructed terrestrial water storage changes from 1901 – present. The data product is created by developing a model that is trained to ∼15 years of GRACE observations. The model takes as input precipitation and temperature information from various models. The authors present a total of six different reconstructions (using two different GRACE data products and three different precipitation data products), with 100 ensemble members for each, to capture the uncertainty associated with each reconstruction. The authors perform various comparisons with other data (sea level budget, streamflow data, BSWB data, and GRACE data itself, as an effort to do validation on their dataset(s), as well as hydrological models. The authors find that by large, their

reconstruction(s) agree better or on par with the hydrology models for the various comparisons (validations) made in the paper.

In general, the paper is thoughtful, well-written, and a welcome addition to the literature. The dataset(s) presented are well-validated by the authors (to the extent possible), and could be very useful for other Earth System studies. I congratulate the authors for this nice contribution.

My primary criticism of the paper is the choice to use the JPL RL05 data rather than the RL06 data (released in October 2018), primarily due to timeliness. Understandably, much of the analysis was likely done prior to the release of the RL06 data, and it would require substantial efforts to redo the analysis. The authors did show that the reconstructions were much more sensitive to the choice of precipitation dataset than the GRACE data, so it is entirely plausible that calibrating the model to RL06 data would make little difference in the results. The hesitation comes with an anticipated use-case of the dataset, as mentioned by the authors (abstract and introduction), which is to fill the gap in between GRACE and GRACE-FO and to "reconcile" the two datasets. The first GRACE-FO data will be in so-called "RL06" data standards. It would behoove the authors to address this discrepancy, and provide some analysis/insights on whether any conclusions change when using RL06 data to calibrate the model. The authors discuss the potential for errors in low degree spherical harmonics (Section 4.3), and in fact, many of the changes from RL05 to RL06 occur in the low degree harmonic coefficients for the JPL data product, including the "mean pole correction" of the C21/S21 coefficient as recommended by Wahr et al., 2015.

Specific Comments:

Section 2.3.2: The model is calibrated independently for each mascon. It is unclear to me – does this mean for the JPL data product it is done on each 3-degree mascon, while on the GSFC data product it is done on each 1-degree mascon? There are many more mascons in the GSFC data product than degrees of freedom in the

GRACE dataset – but perhaps this does not matter for the model calibration since spatial correlations are taken into account. Can you comment?

Figure 4c and 4d: It is unclear to me what each data point represents. Is each dot for a single mascon?

Section 3.4: The title "Global Average" is perhaps misleading since it does not include ocean areas. Suggested revision.

Figure 7: Are these simply the global average (area weighted) of Figure 5 and 6?

Section 4.3: This analysis is done excluding Greenland and Antarctica. Are Greenland and Antarctica excluded from the actual GRACE data (JPL and GSFC) when computing correlations/RMS with altimetry/steric information in Figure 11b/c? I wonder what the impact of including/excluding it is? Presumably small, but some discussion on this would make for a better comparison.

Section 4.3: It is hypothesized that low degree errors could be responsible for the GRACE data having a worse correlation than the modeled data. I agree. I could also envision errors in high degrees also being a culprit. The mascon solutions used in theory do not necessitate any post-processing, but it is very likely that residual longitudinal stripes remain. The GRACE-REC model should not calibrate to these residual stripes, but rather the signal since the stripes are more stochastic in nature from month to month. However, it is plausible that residual stripes could contaminate correlation/RMSE comparison with a detrended/deseasoned timeseries of presumed ocean mass from sea level budget analysis (altimetry/steric).

Section 4.4: Could you include some discussion of the length of the timeseries of the BSWB data? Figure 12 is confusing because in Figure 12a, the BSWB data does not overlap with the GRACE data record. However, Figure 12b/c compare the BSWB data with the actual GRACE data – inherently implying some overlap.

Section 4.4 and 4.5: In both sections it is pointed out there is slightly better performance

in GSFC than JPL, and this is potentially owed to the better spatial resolution of the GSFC data. Did you apply the scale factors to the JPL data? These are designed to reduce such leakage error on the basin scale. If not, I suggest doing so for this analysis. Second, when making these comparisons, is the length of the data record always consistent? The JPL data both begins before, and extends after, the GSFC data. The tails of the GRACE dataset are of worse quality, and I am curious if the inclusion of these extra months on the JPL data is perhaps responsible for the inferior performance.

---

## Referee Comment (RC2) · Anonymous Referee #2 · 25 Mar 2019

In their study the authors use three different precipitation and temperature products to reconstruct past variability of terrestrial water storage (TWS) from 2017 back to 1901. The reconstruction is performed by estimating the parameters of a statistical model which is calibrated by relating precipitation and temperature to observed TWS from the GRACE satellite mission. To account for temporally and spatially correlated errors in the reconstructed TWS the authors apply a spatial autoregressive model to generate a large number of ensemble members representing the uncertainty of the estimated TWS anomalies. Afterwards, the derived reconstructions are evaluated against different independent datasets, showing the value of the dataset for different hydrological and climate applications.

[Figure]

The presented data and method are new and sufficiently described in the text. Long and consistent time series of TWS as presented here will be very useful in future for many different user groups, thus it is a valuable contribution to ESSD.

Generally, the manuscript is well structured and well written. Data access is easy and well documented. Downloaded data are ready to use without problems. The data is of high quality as shown by the authors in several appropriate evaluations.

General comments:

Chapter 2.2: Instead of ERA-Interim as used in the study, it would be better to use the new ERA5 reanalysis (at least for the next update of the reconstruction, as ERA-Interim production will eventually end). Probably this would even improve the quality of the reconstruction.

Chapter 2.3: Some aspects of the modelling approach are unclear to me:

Where does Eq. 5 come from? A sentence on this for explanation would be helpful for the reader.

Does time t in Eq. 6 refer to months and TWS(t) to a monthly average (in contrast to before, where t was time in days)? If so, the notation should be adjusted accordingly, e.g. using t' and mean(TWS) to distinguish monthly from daily resolution. $\varepsilon$ also depends on (monthly) t, this should be indicated in Eq. 6 (and accordingly in Eq. 8), e.g. with $\varepsilon\_t'$.

Chapter 2.4.2:

I do not understand Eq. 13: To my understanding $\sigma\_\eta$ is the "variance of the autoregressive process" (line 8) which should be "larger than that of the driving white noise process" (line 9), which is $\sigma\_\varepsilon$. However, for large autocorrelation $\varphi$ the expression $\sqrt{(1-\varphi^2)}$ approaches zero, thus $\sigma\_\eta$ is smaller than $\sigma\_\varepsilon$ for any autocorrelation different from zero. Please comment on this.

Specific comments:

P. 5, line 9: (typo) adjustement must be adjustment

P. 9, line 20: (Eq. 8) dependence on time for GRACEREC and $\varepsilon$ should be visible in equation.

P. 12, line 9: does "ensemble hindcast" refer to a mean of all 6 reconstructions (each with 100 ensemble members)? Please point this out more clearly. Otherwise, please indicate which reconstruction is evaluated.

P. 13, line 19: so no SAR model was used for daily products? Maybe mention this and the reason for it explicitly.

P. 15, line 13ff: Did you evaluate the difference between the two GRACE solutions in advance? Usually, GRACE solutions of different processing centers do not differ largely, thus it is not surprising that they lead to similar reconstructions.

P. 16, line 19ff: This is a repetition of P. 14, line 10-13. It should be summarized and discussed at one location.

P. 17, line 5: The GRACE solution from Graz is officially called ITSG-Grace2018 (not just ITSG2018). Mayer-Gürr et al., 2016 is an outdated reference; if you used the 2018 solution, please cite: Mayer-Gürr, Torsten; Behzadpur, Saniya; Ellmer, Matthias; Kvas, Andreas; Klinger, Beate; Strasser, Sebastian; Zehentner, Norbert (2018): ITSG-Grace2018 - Monthly, Daily and Static Gravity Field Solutions from GRACE. GFZ Data Services. http://doi.org/10.5880/ICGEM.2018.003

P. 19, line 8f: Please comment on how this is possible since GRACE cannot resolve features as small as $1°$.

P. 19, line 19: "size smaller than . . ." Do you mean "size larger than. . ."? Otherwise I do not understand why you only use the very small basins.

P. 19, line 20: "leaving 12'496 stations", please indicate number of stations for each

time period, as in Figure 13c only 9306 stations are evaluated.

Figure 1b: y-axis label should be changed from cm H2O to TWS [cm]

Figure 3 caption, line 2: delete "also"

Figure 4: a, b and e are too small. In c, only one x-axis label is printed, please add more.

Figure 7: Please mention to what the bars and lines refer to. Standard deviation, min and max? Is the global mean computed with or without Greenland and Antarctica?

Figure 8: In 8a for some time series (red, purple, light blue) the numbers at the scale are missing. b and c are too small to distinguish different reconstructions.

Figure 13d: Repetition of legend from 13b would be nice, to see at a glance what is displayed here.

---

## Referee Comment (RC3) · Anonymous Referee #3 · 27 Mar 2019

General Comments:

The paper "GRACE-GEC: a reconstruction of climate-driven water storage changes of the last century" presents a set of statistical models of TWS trained to two GRACE mascon solutions using multiple precipitation and temperature forcing inputs. The discussion in the paper well framed, providing a detailed methodology and explanation of relevant key decisions in developing that methodology. The paper then provides a product description and evaluation that conveys the information content of the developed models and provides an analysis of that content in a straightforward and logical way. The paper itself is well written, and I did not find any typographical errors or major

grammatical issues anywhere. The level of detail is such that anyone generally familiar with the subject matter can nicely comprehend the discussed work and outcomes. As a whole, I believe that this paper is very close to a final form, and primarily have clarification questions and small probing questions that I would like to possibly see further discussed.

Data access was straightforward and is well documented. My only suggestion would be to have a more meaningful naming schema for the zip files. For example, a name that tells me "trained with JPL, forced by MSWEP, spanning 1979-2016", as in the names of the NetCDFs themselves, rather than requiring that I refer to the README for that information. As for ease of use, I was able to create a Jupyter Notebook with Python 3.7 in under two minutes that already had me using the data. The choice of NetCDF is very much appreciated.

As a general comment, with the release of JPL's RL06 Mascons, do you plan to update the JPL-trained models? Or perhaps more generally, is there a plan in place to continually produce new models when new GRACE solutions are available for training? Similarly, when GRACE-FO is operational, what plan is in place to extend the training datasets with new months? Will this be continually re-done, or is there even a benefit to doing a new run with each new month? A general comment in the paper discussing the sensitivity of the models to additional months of GRACE forcing would be appreciated.

—

Specific Comments:

- p. 5 line 5 / Table 1 - Did you consider using formulations of the two mascon solutions that have equivalent GIA models removed? For example, on looking at the GSFC mascon website, those mascons are distributed with either the A et al. model or ICE-6G model removed. You could compute consistent reconstructions for both mascon sets but using consistent GIA models. Also, does any of this matter since you are using a detrended dataset for the training of your reconstructions? This should probably be

clarified.

- p. 5 line 5 / Table 1 - For JPL, why have you selected the CRI filtered solution and what considerations must be made as a result of that choice? Are you using that solution at it's gridded resolution (0.5 degree x 0.5 degree) or on a mascon-by-mascon basis (4551 mascons). If at the gridded level, are you forcing reconstruction outputs to be equal over all grid cells in each mascons or allowing for spatial variation within individual mascons? Same question for the temperature and precip inputs over these mascons? Also, are there any other differences between the mascons that are important to consider (or alternatively, is this even in the scope of your paper)?

- Section 2.1 - Relating to the last two questions, do you handle each solution at their own native resolutions or are they placed onto a common grid? It appears that the model outputs from the GSFC-driven runs were placed onto a half-degree grid. How were they handled in the training portion of the products developments?

- p. 8 line 9 - Why is the seasonal cycle removed prior to the calibration step? What are the repercussions of this decision on the reconstruction? This is somewhat addressed in Section 3 but at the time is a major open question to the reader.

- p. 8 line 20 - In your discussion of error sources, how do spatially correlated errors in the GRACE solutions impact the work? You have "mascon binned" your reconstruction, so to speak, but the GRACE mascons themselves are not independent mass estimates (especially in the case of the 1-arc-degree GSFC mascons). This bias error source is in addition to the measurement errors from GRACE and is difficult to address. Have you included anything to account for this?

- p. 16 line 18-22 - This seems redundant with section 3.5.

- p. 16 line 23-p. 17 line 2 - It is unclear if/why this is unexpected. If training was done at the mascons scale, it would seem that larger scales aggregating multiple mascons would show well calibrated agreement se a necessary but not sufficient condition on

the dataset.

- Section 4.2 - In addition to the lower spatial resolution, the Kalman smoothed daily GRACE solution is correlated in time; is your comparison to the GRACE-REC products at all different than for the monthly solutions as a result of this?

- Figure 7 - The dark/light distinction could be a little more obvious, rather than having to read deeply into the caption, and also have a stronger contrast.

- p. 19 line 8 - GSFC mascons are smaller, yes, but does the GSFC solution actually have better resolution than the JPL mascons? This is related to the comments about how the JPL mascons are handled and how spatial correlations in the solutions are handled (ex: higher cross-mascon correlations in the GSFC solution than with JPL due to the smaller mascon sizes).

- In the abstract, possible user groups and applications were identified. Would an example of the application of this work in one of those areas be within the scope of this paper? Also, if the reconstruction is based on de-seasoned and de-trended GRACE information, is bridging the GRACE/GRACE-FO gap actually an application? What limitations are placed on such a use?

---

## Author Comment (AC1) · 3 Jun 2019

We wish to thank all three anonymous referees for their constructive feedback, thoughtful comments and suggestions.

Please refer to the attached zip file, containing:

1) response letter to referees 2) revised manuscript with tracked changes 3) revised manuscript 4) supplementary information

Please also note the supplement to this comment:

[Figure]

https://www.earth-syst-sci-data-discuss.net/essd-2019-25/essd-2019-25-AC1-supplement.zip

---

## Author Response (AR1)

**This document contains:**
1. Detailed response to referees' comments and suggestions
2. List of minor updates in data analysis
3. List of minor updates in data presentation

**1. Detailed response to referees' comments and suggestions**

**Referee #1 (Remarks to the Author):**

In general, the paper is thoughtful, well-written, and a welcome addition to the literature.The dataset(s) presented are well-validated by the authors (to the extent possible), andcould be very useful for other Earth System studies. I congratulate the authors for thisnice contribution.

> We thank the referee for reviewing our work and for the positive evaluation of the study. Please find our point-by-point response below.

My primary criticism of the paper is the choice to use the JPL RL05 data rather thanthe RL06 data (released in October 2018), primarily due to timeliness. Understand-ably, much of the analysis was likely done prior to the release of the RL06 data, and itwould require substantial efforts to redo the analysis. The authors did show that the re-constructions were much more sensitive to the choice of precipitation dataset than theGRACE data, so it is entirely plausible that calibrating the model to RL06 data wouldmake little difference in the results. The hesitation comes with an anticipated use-caseof the dataset, as mentioned by the authors (abstract and introduction), which is to fillthe gap in between GRACE and GRACE-FO and to "reconcile" the two datasets. Thefirst GRACE-FO data will be in so-called "RL06" data standards. It would behoove theauthors to address this discrepancy, and provide some analysis/insights on whetherany conclusions change when using RL06 data to calibrate the model. The authorsdiscuss the potential for errors in low degree spherical harmonics (Section 4.3), and infact, many of the changes from RL05 to RL06 occur in the low degree harmonic coef-ficients for the JPL data product, including the "mean pole correction" of the C21/S21coefficient as recommended by Wahr et al., 2015.

> Thank you very much for this suggestion. In our revised version of the manuscript, we use RL06 instead of RL05. While this did not massively improve or change the reconstructions, it ensures future consistency with the first GRACE-FO data from JPL.

Specific Comments:

Section 2.3.2: The model is calibrated independently for each mascon. It is unclearto me – does this mean for the JPL data product it is done on each 3-degree mas-con,

while on the GSFC data product it is done on each 1-degree mascon? There are many more mascons in the GSFC data product than degrees of freedom in the GRACE dataset – but perhaps this does not matter for the model calibration since spatial correlations are taken into account. Can you comment?

Yes, as mentioned in section 2.3.1, the calibration is conducted at each mascon (3° for JPL and 1° for GSFC). Because GRACE effective resolution is lower than 1°, neighboring GSFC mascons essentially represent the same signal as you mentioned. However, this does not really matter for the model calibration and we found no indication that calibrating the model at GSFC resolution (1°) leads to overfitting or unreasonable parameter values. In fact, the MCMC algorithm does not provide only one model parameter set at a given mascon, but a (more robust) distribution of acceptable model parameter sets. These parameter distributions do exhibit spatial auto-correlation, reflecting the spatial "smoothness" or oversampling inherent to the GSFC solution.

Figure 4c and 4d: It is unclear to me what each data point represents. Is each dot for a single mascon?

Yes, this has been made clearer in the legend of Figure 4.

Section 3.4: The title "Global Average" is perhaps misleading since it does not include ocean areas. Suggested revision.

Title changed to "Global land averages"

Figure 7: Are these simply the global average (area weighted) of Figure 5 and 6?

Thank you very much for this question. No, Figure 7 depicts box and whisker plots of the values shown in Figure 5 and 6 (we follow the general convention of $10^{th}$, $25^{th}$, $50^{th}$, $75^{th}$, and $90^{th}$ percentiles). Note that the calculation of the percentiles takes into account mascon area (coastal JPL mascons that have a smaller area have a smaller weight). This has now been made clearer in the legend of Figure 7, as well as of Figures 10, 12 and 13 which have similar representations.

Section 4.3: This analysis is done excluding Greenland and Antarctica. Are Greenland and Antarctica excluded from the actual GRACE data (JPL and GSFC) when computing correlations/RMS with altimetry/steric information in Figure 11b/c? I wonder what the impact of including/excluding it is? Presumably small, but some discussion on this would make for a better comparison.

Yes, Greenland and Antarctica are excluded for all products, mainly because WRR models are not meant to be used in those regions and thus either do not provide output or produce spurious values in those regions. We checked this and found that including Greenland and Antarctica only has a small effect. The agreement with altimetry-steric decreases for some

WRR models and for JPL mascons and slightly increases for GSFC mascons and some of the reconstructions. Because this additional analysis (including Greenland and Antarctica) does not provide a fair basis for comparing all the different products (because of the WRR models), we prefer to exclude it. We note that global means both excluding or including Greenland and Antarctica are also readily available as part of the final product.

Section 4.3: It is hypothesized that low degree errors could be responsible for theGRACE data having a worse correlation than the modeled data. I agree. I could alsoenvision errors in high degrees also being a culprit. The mascon solutions used intheory do not necessitate any post-processing, but it is very likely that residual longi-tudinal stripes remain. The GRACE-REC model should not calibrate to these resid-ual stripes, but rather the signal since the stripes are more stochastic in nature frommonth to month. However, it is plausible that residual stripes could contaminate corre-lation/RMSE comparison with a detrended/deseasoned timeseries of presumed oceanmass from sea level budget analysis (altimetry/steric).

Thank you for this comment, we were not aware of this possibility. This has been included in the discussion: *"... (e.g. caused by errors in low degree spherical harmonics or residual longitudinal stripes)…"*.

Section 4.4: Could you include some discussion of the length of the timeseries of theBSWB data? Figure 12 is confusing because in Figure 12a, the BSWB data does notoverlap with the GRACE data record. However, Figure 12b/c compare the BSWB datawith the actual GRACE data – inherently implying some overlap.

Thank you for this remark. The BSWB data in theory covers the period 1979-2015, however, calculation of the basin-scale water balance is also subject to availability of runoff measurements which varies a lot depending on the basin. Thus, while BSWB data shown in Figure 12a is not available after 1998, many stations do overlap with the GRACE period. Our intention for selecting the Ob basin in Figure 12a was also to illustrate how the reconstruction can reconcile gaps between datasets from multiple sources.

This has been made clearer in the text: *"The temporal coverage of BSWB estimates at each river basin thus depends on the availability of runoff data and does not always cover the GRACE time period."*

Section 4.4 and 4.5: In both sections it is pointed out there is slightly better performance in GSFC than JPL, and this is potentially owed to the better spatial resolution of theGSFC data. Did you apply the scale factors to the JPL data? These are designedto reduce such leakage error on the basin scale. If not, I suggest doing so for thisanalysis. Second, when making these comparisons, is the length of the data record always consistent? The JPL data both begins before, and extends after, the GSFCdata. The tails of the GRACE dataset are of worse quality, and I am curious if the inclusion of these extra months on the JPL data is perhaps responsible for the inferior performance.

Thank you for this comment. We note that this point is only valid for section 4.4 as GRACE data is not used in section 4.5. It is true that the CLM4-based scale factors could be applied to JPL data when recovering the basin averages used for the analysis in section 4.4. We now apply the scale factors for this analysis (now noted in the figure legend). Also we made sure that both JPL and GSFC are evaluated over the same time period.

We also rephrased the sentence to make clear that our intention here is to explain why GSFC-based products seem to have better performance than the JPL-based products in sections 4.4. and 4.5. We have added the following clarification: *"This mainly occurs because the meteorological forcing is aggregated at a resolution of 1° in the case of GSFC-based products, allowing the GSFC reconstructions to provide a slightly more localized signal."*

**Referee #2 (Remarks to the Author):**

In their study the authors use three different precipitation and temperature products toreconstruct past variability of terrestrial water storage (TWS) from 2017 back to 1901.The reconstruction is performed by estimating the parameters of a statistical modelwhich is calibrated by relating precipitation and temperature to observed TWS from theGRACE satellite mission. To account for temporally and spatially correlated errors inthe reconstructed TWS the authors apply a spatial autoregressive model to generatea large number of ensemble members representing the uncertainty of the estimatedTWS anomalies. Afterwards, the derived reconstructions are evaluated against differ-ent independent datasets, showing the value of the dataset for different hydrological and climate applications.

The presented data and method are new and sufficiently described in the text. Longand consistent time series of TWS as presented here will be very useful in future formany different user groups, thus it is a valuable contribution to ESSD.

Generally, the manuscript is well structured and well written. Data access is easy andwell documented. Downloaded data are ready to use without problems. The data is ofhigh quality as shown by the authors in several appropriate evaluations.

We thank the referee for reviewing our work and for the positive evaluation of the study. Please find our point-by-point response below.

General comments:

Chapter 2.2: Instead of ERA-Interim as used in the study, it would be better to usethe new ERA5 reanalysis (at least for the next update of the reconstruction, as ERA-Interim production will eventually end). Probably this would even improve the quality ofthe reconstruction.

Thank you for this suggestion. In our revised version of the manuscript, we use the newly available ERA5 instead of ERA-Interim. We confirm that the quality of the ERA5-driven reconstruction improved very much as a result of this change. ERA-Interim-based products often had the lowest performance among all reconstructions, but as a result of the update, ERA5-based reconstructions now often yield the best performance. Figure legends and in-text discussions have been updated where necessary.

Chapter 2.3: Some aspects of the modelling approach are unclear to me: Where does Eq. 5 come from? A sentence on this for explanation would be helpful forthe reader.

Thank you for this comment. We realize that this was not entirely clear. We have a made a minor adjustment to Equation 1, which now leads Equation 5 to be more intuitive. In practice, this modification does not change the reconstructed signals. The full development of how Equation 5 is obtained is also provided and illustrated in the Supplementary Material.

Equation 5 is also better explained in the main text: "*The initial value of the storage is thus obtained as the ratio between the rate of water input and the rate of water loss (also see the full development in Supplementary Information)*"

Does time t in Eq. 6 refer to months and TWS(t) to a monthly average (in contrast tobefore, where t was time in days)? If so, the notation should be adjusted accordingly,e.g. using t' and mean(TWS) to distinguish monthly from daily resolution.εalso de-pends on (monthly) t, this should be indicated in Eq. 6 (and accordingly in Eq. 8), e.g.withε_t'.

Thank you for noting this. We have replaced $t$ with $t_m$ whenever we referred to monthly resolution. Equations in the remainder of the manuscript have been updated accordingly.

Chapter 2.4.2:I do not understand Eq. 13: To my understanding σ_η is the "variance of the autoregressive process" (line 8) which should be "larger than that of the driving white noise process" (line 9), which is σ_ε. However, for large autocorrelation φ the expression $\sqrt{1-\varphi^2}$ approaches zero, thus σ_η is smaller than σ_ε for any autocorrelation differ-ent from zero. Please comment on this.

Thank you for your question. There was apparently some confusion, $\sigma_\eta$ is the variance of the noise process and $\sigma_\varepsilon$ is the variance of the auto-regressive process, not the other way around. Taking this into account, your interpretation of the equation is entirely correct. This was made clearer in the text: "*This accounts for the fact that the variance of an autoregressive process ($\boldsymbol{\sigma_\varepsilon}$) is larger than that of the driving white noise process ($\boldsymbol{\sigma_\eta}$).*"

Specific comments:

P. 5, line 9: (typo) adjustement must be adjustment

Corrected, thank you.

P. 9, line 20: (Eq. 8) dependence on time for GRACEREC andεshould be visible inequation.

Corrected, thank you.

P. 12, line 9: does "ensemble hindcast" refer to a mean of all 6 reconstructions (eachwith 100 ensemble members)? Please point this out more clearly. Otherwise, pleaseindicate which reconstruction is evaluated.

Thanks for this comment. This evaluation is for the 100 ensemble members of the JPL-MSWEP reconstruction. This is now indicated in the caption.

P. 13, line 19: so no SAR model was used for daily products? Maybe mention this andthe reason for it explicitly.

Thank you for noting this. Yes, the reason is that calibrating a robust SAR model for the daily resolution is impossible since GRACE observations are at monthly resolution. This was added to the main text: *"The reason for this is that no SAR model (Section 2.4.2) can be reliably calibrated at the daily resolution as the two training GRACE datasets have monthly resolution"*

P. 15, line 13ff: Did you evaluate the difference between the two GRACE solutions in advance? Usually, GRACE solutions of different processing centers do not differlargely, thus it is not surprising that they lead to similar reconstructions.

We agree that this is not too surprising, however, because we get this question a lot, this is why we conducted this assessment.

P. 16, line 19ff: This is a repetition of P. 14, line 10-13. It should be summarized anddiscussed at one location.

Thank you, this was corrected.

P. 17, line 5: The GRACE solution from Graz is officially called ITSG-Grace2018 (notjust ITSG2018). Mayer-Gürr et al., 2016 is an outdated reference; if you used the2018 solution, please cite: Mayer-Gürr, Torsten; Behzadpur, Saniya; Ellmer, Matthias;Kvas, Andreas; Klinger, Beate; Strasser, Sebastian; Zehentner, Norbert (2018): ITSG-Grace2018 - Monthly, Daily and Static Gravity Field Solutions from GRACE. GFZ DataServices. http://doi.org/10.5880/ICGEM.2018.003

Thank you, we have updated the reference and figure legends accordingly.

P. 19, line 8f: Please comment on how this is possible since GRACE cannot resolvefeatures as small as 1◦.

We agree that the wording was inadequate. We have replaced "the higher spatial *resolution* of the GSFC mascons" with "the higher spatial *sampling* of the GSFC mascons".

We also rephrased the sentence to make clear that our intention here is to explain why GSFC-based products seem to have better performance than the JPL-based products in sections 4.4. and 4.5. We have added the following explanation: "*This mainly occurs because the meteorological forcing is aggregated at a resolution of 1° in the case of GSFC-based products, allowing the reconstruction to provide a slightly more localized signal.*"

P. 19, line 19: "size smaller than..." Do you mean "size larger than..."? Otherwise Ido not understand why you only use the very small basins.

Thank you for noting this. Here, we focus on basins that are small enough to completely fall within the footprint of a GRACE mascon or a WRR2 grid cell. The main reason for this is that the number of large basins available prior to 1980 is extremely small compared to the thousands of measurements made at small basins back until 1901 and before. We are aware that large-scale mass changes are not necessarily representing the dynamics of such small catchments. However, the purpose is not to obtain a perfect match, but to diagnose potential relative changes over time in the performance of the century-long reconstruction. We have added the following explanation in the main text:

"*The reason for focusing on small basins is that a much larger number of them is available in the early century (compared to the number of large basins, which are the focus of section 4.4). We note that the unavoidable mismatch between large-scale mass changes and local catchment runoff dynamics is to some extent alleviated by the spatial coherence of anomalies in weather patterns at yearly scale.*"

P. 19, line 20: "leaving 12'496 stations", please indicate number of stations for each time period, as in Figure 13c only 9306 stations are evaluated.

Thank you for noting this. This is now indicated in the legend: "*(n=1274, 8065 and 9306 for 1901-40, 1941-80 and 1981-2010 respectively).*"

Figure 1b: y-axis label should be changed from cm H2O to TWS [cm]

Corrected.

Figure 3 caption, line 2: delete "also"

Corrected.

Figure 4: a, b and e are too small. In c, only one x-axis label is printed, please addmore.

Corrected.

Figure 7: Please mention to what the bars and lines refer to. Standard deviation, minand max? Is the global mean computed with or without Greenland and Antarctica?

Thank you for this feedback, we agree that the legend needed more clarity. Figure 7 depicts box and whisker plots of the values shown in Figure 5 and 6 (we follow the general convention of $10^{th}$, $25^{th}$, $50^{th}$, $75^{th}$, and $90^{th}$ percentiles). Note that the calculation of the percentiles takes into account mascon area (as coastal JPL mascons can have a smaller area). This has been made clearer in the legend of Figure 7, as well as of Figures 10, 12 and 13 which have similar representations. Greenland and Antarctica are always excluded from these figures.

Figure 8: In 8a for some time series (red, purple, light blue) the numbers at the scaleare missing. b and c are too small to distinguish different reconstructions.

The missing numbers were added in 8a. With respect to 8b and 8c, the fact that the different reconstructions are difficult to distinguish in terms of inter-annual variability (over the GRACE time period) is actually the correct interpretation of this figure. We have made this clearer in the text and note that the different reconstructions can also be better distinguished in 8a.

Figure 13d: Repetition of legend from 13b would be nice, to see at a glance what isdisplayed here.

Corrected, thank you.

**Referee #3 (Remarks to the Author):**

General Comments: The paper "GRACE-GEC: a reconstruction of climate-driven water storage changes ofthe last century" presents a set of statistical models of TWS trained to two GRACEmascon solutions using multiple precipitation and temperature forcing inputs. The dis-cussion in the paper well framed, providing a detailed methodology and explanationof relevant key decisions in developing that methodology. The paper then provides aproduct description and evaluation that conveys the information content of the devel-oped models and provides an analysis of that content in a straightforward and logicalway. The paper itself is well written, and I

did not find any typographical errors or major grammatical issues anywhere. The level of detail is such that anyone generally familiarwith the subject matter can nicely comprehend the discussed work and outcomes. Asa whole, I believe that this paper is very close to a final form, and primarily have clari-fication questions and small probing questions that I would like to possibly see furtherdiscussed.

We thank the referee for reviewing our work and for the positive evaluation of the study. Please find our point-by-point response below.

Data access was straightforward and is well documented. My only suggestion would beto have a more meaningful naming schema for the zip files. For example, a name thattells me "trained with JPL, forced by MSWEP, spanning 1979-2016", as in the namesof the NetCDFs themselves, rather than requiring that I refer to the README for thatinformation.

Thanks for this comment. We have made the .zip file names more meaningful.

As for ease of use, I was able to create a Jupyter Notebook with Python3.7 in under two minutes that already had me using the data. The choice of NetCDF isvery much appreciated.

As a general comment, with the release of JPL's RL06 Mascons, do you plan to updatethe JPL-trained models? Or perhaps more generally, is there a plan in place to con-tinually produce new models when new GRACE solutions are available for training?

In response to a suggestion by another referee, we have updated the JPL dataset used here to RL06. As discussed in section 4.1.1, we find that the reconstructions are not very sensitive to the employed GRACE training dataset. We would update the model if 1) there is a major breakthrough in GRACE processing technique or 2) we find a significantly improved and still as simple formulation of TWS changes (i.e. Eq. 1).

Similarly, when GRACE-FO is operational, what plan is in place to extend the trainingdatasets with new months? Will this be continually re-done, or is there even a benefit todoing a new run with each new month?

With the exception of the two cases mentioned just above, the plan is to update the ERA-5 version on a yearly basis (or occasionally more frequently upon reasonable request), provided the corresponding author is able to secure both funding and time for making these updates.

A general comment in the paper discussing thesensitivity of the models to additional months of GRACE forcing would be appreciated.

Thanks for this comment. This has been added in the main text: *"[...] updates of the two reconstructions driven by ERA5 will be published when needed. We note that because including additional GRACE months only barely improves the quality of the model fit, no systematic re-calibration of the models is planned at this stage."*

Specific Comments:

- p. 5 line 5 / Table 1 - Did you consider using formulations of the two mascon solutionsthat have equivalent GIA models removed? For example, on looking at the GSFCmascon website, those mascons are distributed with either the A et al. model or ICE-6G model removed. You could compute consistent reconstructions for both masconsets but using consistent GIA models. Also, does any of this matter since you are usinga detrended dataset for the training of your reconstructions? This should probably be clarified.

Thanks for this comment. Yes, it actually does not really matter since the detrended dataset is used during model training. This has been clarified in section 2.3.2: *"We note that as a result, the choice of the GIA model used in GRACE processing (Table 1) does not impact the model calibration"*

- p. 5 line 5 / Table 1 - For JPL, why have you selected the CRI filtered solutionand what considerations must be made as a result of that choice? Are you usingthat solution at it's gridded resolution (0.5 degree x 0.5 degree) or on a mascon-by-mascon basis (4551 mascons). If at the gridded level, are you forcing reconstructionoutputs to be equal over all grid cells in each mascons or allowing for spatial variationwithin individual mascons? Same question for the temperature and precip inputs overthese mascons? Also, are there any other differences between the mascons that areimportant to consider (or alternatively, is this even in the scope of your paper)?

Thank you for this comment. The CRI filtered solutions are recommended by JPL for land hydrology analyses. The meteorological forcings are averaged over the footprint of the land part of the mascons. This has been made clearer in the text of section 2.3.1: "*The meteorological forcing is always spatially averaged over the spatial footprint of the GRACE mascons.*".

Training is always done at mascon-scale (mascon-by-mascon basis) as mentioned in section 2.3.1. Concerning the differences in terms of the processing of these solutions, they certainly exist and the methodologies are well described by the cited references. We do not extensively discuss these differences here as 1) this would be outside the scope of the paper, and 2) the choice of the training GRACE dataset was found to be of secondary importance (as shown in section 4.1.1), so that, even if we would include such a discussion, it would not really aid the interpretation of our results.

- Section 2.1 - Relating to the last two questions, do you handle each solution at

theirown native resolutions or are they placed onto a common grid? It appears that themodel outputs from the GSFC-driven runs were placed onto a half-degree grid. Howwere they handled in the training portion of the products developments?

Thanks for this comment. As mentioned above and in section 2.3.1, all model training and model output is handled at the mascon level. Final products are provided on a half-degree grid as this seems to be the most convenient for most users.

- p. 8 line 9 - Why is the seasonal cycle removed prior to the calibration step? What arethe repercussions of this decision on the reconstruction? This is somewhat addressedin Section 3 but at the time is a major open question to the reader.

Thank you for this comment. Removing the seasonal cycle allows us to focus the model on those deviations from typical TWS variability that are hard to predict (while seasonality is easily defined from GRACE data alone). This is now better explained in section 2.3.2: "*Removing the seasonal cycle lets the model calibration focus on capturing the inter-annual variability correctly*".

This has little repercussion on the reconstruction, except that the reconstruction likely cannot be used investigate long-term changes in seasonality as mentioned in section 3.1.

- p. 8 line 20 - In your discussion of error sources, how do spatially correlated errors inthe GRACE solutions impact the work? You have "mascon binned" your reconstruction,so to speak, but the GRACE mascons themselves are not independent mass estimates(especially in the case of the 1-arc-degree GSFC mascons). This bias error source isin addition to the measurement errors from GRACE and is difficult to address. Haveyou included anything to account for this?

Thanks for this comment. This is true, we implicitly include this type of errors in the SAR model. This is now more clearly mentioned in section 2.4.1: "*They include measurement and leakage errors from GRACE*".

As mentioned by the referee, spatially correlated errors in GRACE arise for a variety of reasons, and are difficult to address and to isolate. In our case, the SAR model can only provide a bulk representation of the spatial-temporal structure and magnitude of these errors. Our intention is to provide an overall estimate of the expected mismatch between the reconstruction and GRACE data (a mismatch caused by a wide variety of factors, including the interdependence of neighboring mascons). Our goal with the ensemble members is that this error estimate will also be easily computed when the end user wants to perform spatial and/or temporal aggregation.

- p. 16 line 18-22 - This seems redundant with section 3.5.

Thank you, this has been corrected.

- p. 16 line 23-p. 17 line 2 - It is unclear if/why this is unexpected. If training was doneat the mascons scale, it would seem that larger scales aggregating multiple masconswould show well calibrated agreement se a necessary but not sufficient condition on the dataset.

> We were not entirely sure how to interpret/understand this comment. We mean that calibrating the relationships locally does not automatically ensure that the global averages will also match. For instance, having poor model skill over several key regions (see e.g. Figures 5 and 6) could have contaminated the global averages, but this is not the case here.

- Section 4.2 - In addition to the lower spatial resolution, the Kalman smoothed dailyGRACE solution is correlated in time; is your comparison to the GRACE-REC productsat all different than for the monthly solutions as a result of this?

> Thank you for this comment. The actual (true) TWS itself can be expected to be highly auto-correlated, especially at the daily scale, however, it is also true that the Kalman smoothing could further increase the autocorrelation of the time series. This is now mentioned in the text: *"(note that the solution is also correlated in time as a result of the Kalman smoothing)"*.
>
> While additional smoothing likely negatively affects all skill scores shown in Fig.10bc, we do not think that this would bias the comparison between WRR2 and GRACE-REC products (none of the two should be more affected than the other by this issue).

- Figure 7 - The dark/light distinction could be a little more obvious, rather than havingto read deeply into the caption, and also have a stronger contrast.

> We have made the distinction more obvious by enhancing the contrast and have added a legend in the figure.

- p. 19 line 8 - GSFC mascons are smaller, yes, but does the GSFC solution actuallyhave better resolution than the JPL mascons? This is related to the comments abouthow the JPL mascons are handled and how spatial correlations in the solutions arehandled (ex: higher cross-mascon correlations in the GSFC solution than with JPL dueto the smaller mascon sizes).

> Thank you for this comment. Our understanding is that both solutions have approximately the same effective spatial resolution. What we mean here is that, because the meteorological forcing is aggregated over a smaller footprint in the case of GSFC, the GSFC reconstructions occasionally provide a more localized estimate of TWS changes. We do not mean to say that GRACE GSFC has higher resolution than GRACE JPL.
>
> This has been made clearer in the text, also in response to a previous comment from another referee: *"This mainly occurs because the meteorological forcing is aggregated at*

*a resolution of 1° in the case of GSFC-based products, allowing the reconstruction to provide a slightly more localized signal."*

- In the abstract, possible user groups and applications were identified. Would anexample of the application of this work in one of those areas be within the scope of thispaper? Also, if the reconstruction is based on de-seasoned and de-trended GRACEinformation, is bridging the GRACE/GRACE-FO gap actually an application? Whatlimitations are placed on such a use?

Thank you for these questions. One use case is already implicitly illustrated with the sea level budget in section 4.3. In fact, over 1993-2002, Figure 11a provides a reconstruction-based estimate of the inter-annual variability in the steric contribution. Benchmarking of global hydrological models is also implicitly included in Figures 7, 9, 10, 11, 12 and 13. The first publication describing this type of approach (Humphrey et al. 2017) provides an example application relating to estimating groundwater depletion and Figure 1 in Humphrey et al. 2018 also contains an example of inter-disciplinary application.

As for bridging GRACE/GRACE-FO, we agree that this paper only potentially resolves the question of the inter-annual variability. This should be seen as preparatory work. Our opinion is that the seasonal cycle estimated from GRACE could be in theory extended to cover the data gap without major issues from a climatological point of view (if GRACE and GRACE/FO happened to largely diverge in terms of seasonality, this would rather indicate a problem with the geodesy). With respect to the trends, we anticipate that they could be relatively safely extrapolated for the duration of the data gap, however, this would require a more thorough assessment. We would be very interested to follow-up on this particular application as soon as the first GRACE-FO data becomes available.

**2. List of updates in data analysis**

1. JPL RL05 was replaced with JPL RL06
2. ERA-Interim was replaced with ERA5, leading to a significant improvement.
3. Equations (1) and (5) were slightly modified. In practice, this has no impact on the reconstructed signals.
4. For consistency, all models were re-trained and all products were updated in the online data repository. This new version (v3) replaces the previous version (v3beta).

**3. List of changes in data presentation**

1. The development leading to Equation (5) is now explained in a Supplementary Information.
2. For completeness and in response to a user request, we also illustrate the 2003-2014 GRACE trends, reconstructed GRACE-REC trends and WRR2 trends in Supplementary Figures S2-S4.

[revised manuscript text omitted]

**Evaluation of TWS datasets against GRACE JPL and GRACE GSFC**

[Figure]

compared to JPL
compared to GSFC

Figure 7. Global area-weighted box plots of the performance metrics shown in Figures 5 and 6 for
5 GRACE-REC datasets (blue), and comparison with the performance of global hydrological models
participating in the Earth2Observe Water Resources Reanalysis version 2 (WRR2) (orange). Dark colors
indicate the performance obtained when comparing against 3° x 3° JPL Mascons, and against 1° x 1°
GSFC Mascons for light colors. Note: WRR2 models are driven with MSWEP precipitation and all model
outputs are aggregated to the resolution of the corresponding GRACE dataset. Greenland and Antarctica
10 are always excluded.

[Figure]

[Figure]

[Figure]

Figure 8. (a) Global average of TWS anomalies for the 6 GRACE-REC datasets (excluding Greenland and Antarctica) with an artificial vertical offset added for better visual comparison. (b) Comparison of the 3 GRACE-REC datasets calibrated with GRACE JPL against GRACE JPL (de-trended anomalies).
5 (c) Same as (b) but for GRACE GSFC.

[Figure]

[Figure]

[Figure]

□   Against GRACE global mean (JPL)          ×   Against GRACE global mean (GSFC)

Figure 9. Agreement of the global average of different TWS model estimates (from GRACE-REC (blue) and WRR2 models (orange)) with the observed TWS anomalies from JPL (squares) and GSFC (crosses) solutions.

[Figure]

[Figure]

Figure 10. (a) Comparison between the GRACE-REC daily TWS reconstruction (JPL-MSWEP dataset) and the daily GRACE ITSG-Grace2018 solution for the Mississippi basin (focused over the period 2008-

2014 to improve readability of the high-frequency fluctuations). (b-c) Global area-weighted box plots of

5    the performance metrics of the daily TWS datasets when compared with ITSG-Grace2018 at a spatial

resolution of 5°. Note that some WRR2 models are not included because not all water storage variables were available to us at daily frequency. Greenland and Antarctica are excluded.

[Figure]

[Figure]

Figure 11. (a) Comparison of the global mean TWS reconstructed by GRACE-REC (converted to
equivalent mm sea level) against the ocean mass derived from the sea level budget. (b-c) Evaluation of
the ability of various TWS datasets to close the sea level budget (GRACE estimates in green, GRACE-
REC datasets in blue, and WRR2 models in orange).

[Figure]

[Figure]

Figure 12. (a) Comparison between TWS anomalies derived from atmospheric basin-scale water balance (BSWB), GRACE observations (JPL) and the GRACE reconstruction (JPL-MSWEP dataset). (b-c) Global box plots of the agreement between various TWS products and BSWB estimates (based on the performance metrics at 341 large basins). The scale factors were applied to the JPL data for this specific analysis.

[Figure]

[Figure]

Figure 13. (a) Comparison between century-long measurements of streamflow and the TWS anomalies reconstructed at this location (GSFC-GSWP3 dataset). (b) Scatter plot of the data in (a), by time period. (c) Global box plots of the performance of GRACE-REC and WRR2 models when compared with yearly streamflow anomalies. (d) Global box plots of the performance of the JPL-GSWP3 and GSFC-GSWP3 products when compared with yearly streamflow anomalies, by time period (n=1274, 8065 and 9306 for 1901-40, 1941-80 and 1981-2010 respectively).